# Immunotherapies targeting the oncogenic fusion gene CLDN18-ARHGAP in gastric cancer

Yue Wang[1,5], Hanbing Wang[1,5], Tao Shi [ID][1,5], Xueru Song[1], Xin Zhang[1], Yue Zhang[1], Xuan Wang[2], Keying Che [ID][1], Yuting Luo[1], Lixia Yu[1], Baorui Liu[1] & Jia Wei [ID][1,3,4 ✉]

## Abstract

The CLDN18-ARHGAP fusion gene is an oncogenic driver newly discovered in gastric cancer. It was detected in 9% (8/87) of gastric cancer patients in our center. An immunogenic peptide specifically targeting CLDN18-ARHGAP fusion gene was generated to induce neoantigen-reactive T cells, which was proved to have specific and robust anti-tumor capacity both in in vitro coculture models and in vivo xenograft gastric cancer models. Apart from the immunogenic potential, CLDN18-ARHGAP fusion gene was also found to contribute to immune suppression by inducing a regulatory T (Treg) cell-enriched microenvironment. Mechanistically, gastric cancer cells with CLDN18-ARHGAP fusion activate PI3K/AKT-mTOR-FAS signaling, which enhances free fatty acid production of gastric cancer cells to favor the survival of Treg cells. Furthermore, PI3K inhibition could effectively reverse Treg cells upregulation to enhance anti-tumor cytotoxicity of neoantigen-reactive T cells in vitro and reduce tumor growth in the xenograft gastric cancer model. Our study identified the CLDN18-ARHGAP fusion gene as a critical source of immunogenic neoepitopes, a key regulator of the tumor immune microenvironment, and immunotherapeutic applications specific to this oncogenic fusion.

**Keywords** Gastric Cancer; Fusion Gene; CLDN18-ARHGAP; Immunotherapy; Neoantigen
**Subject Categories** Cancer; Immunology

## Introduction

Gastric cancer (GC) is the world's sixth most diagnosed cancer type and the third leading cause of mortality (Sung et al, 2021), which has been considered as highly heterogeneous both histologically and genetically (Bass et al, 2014; Kim et al, 2018; Smyth et al, 2020). Despite the clinical progress of anti-HER2 targeted therapies and anti-PD-1/PD-L1 immunotherapies (Bang et al, 2010; Janjigian et al, 2021; Kang et al, 2017), specific and effective treatment strategies for GC are still lacking.

The promising innovative therapeutics targeting neoantigens, including neoantigen vaccines and neoantigen-reactive T (NRT) cell therapy, have brought revolutionary breakthroughs in a variety of solid tumors (Leko et al, 2020; Tran et al, 2016; Tran et al, 2014; Yarchoan et al, 2017). Multiple murine and human studies have provided evidence for the efficacy of neoantigen-based treatment, among which neoantigens deriving from driver mutations tend to be more tumor-specific, immunogenic, and tumor-presented, which are prone to initiate better immune responses (Lybaert et al, 2023). For example, the Rosenberg group reported in a metastatic colorectal cancer patient that objective regression of lung metastases was observed after infusion of tumor-infiltrating lymphocytes specifically targeting KRAS G12D mutation (Tran et al, 2016). Also, in previous work, we have constructed a driver mutation-derived neoantigen peptide library targeting hotspot mutations commonly shared among multiple refractory cancers, based on which neoantigen vaccines and NRT therapy had considerable and prolonged anti-tumor effects (Chen et al, 2019). However, current neoantigen-based strategies are mostly derived from single-nucleotide variations (SNVs) and insertions or deletions (INDELs) (Smith et al, 2019; Turajlic et al, 2017), while GC lacks hotspot mutation in SNVs or INDELs (Bass et al, 2014; Kwon et al, 2018; Mariette et al, 2019). Therefore, alternative targets for successful applications of neoantigen-based therapies in GC still need further exploration.

Apart from SNVs and INDELs, neoantigens derived from fusion genes formed by chromosome structural variation (SV) tend to have higher immunogenic and therapeutic potential in cancers lacking hotspot mutation in SNVs or INDELs (Wang et al, 2021). Up till now, several studies have reported that neoantigens derived from fusion genes such as BCR–ABL in chronic myeloid leukemia, tumor-specific translocation breakpoints in Ewing sarcoma, and MYB-NFIB in head and neck cancer, can induce robust antigen-specific responses among patient autologous T cells (Mackall et al, 2008; Pinilla-Ibarz et al, 2000; Smith et al, 2019; Yang et al, 2019). Fortunately, the progress of high-throughput sequencing has provided thorough genomic profiles of GC in recent years, significantly contributing to the detection of novel gene alternations (Bass et al, 2014; Chen et al, 2015; Kumar et al, 2022; Wang et al, 2014). In 2014, The Cancer Genome Atlas (TCGA) first

[1]Department of Oncology, Nanjing Drum Tower Hospital, Affiliated Hospital of Medical School, Nanjing University, Nanjing, China. [2]Department of Oncology, Nanjing Drum Tower Hospital Clinical College of Nanjing Medical University, Nanjing, China. [3]Chemistry and Biomedicine Innovation Center (ChemBIC), Nanjing University, Nanjing, China. [4]Engineering Research Center of Protein and Peptide Medicine, Nanjing University, Nanjing, China. [5]These authors contributed equally: Yue Wang, Hanbing Wang, Tao Shi. ✉E-mail: jiawei99@nju.edu.cn

found the CLDN18-ARHGAP fusion gene in 4% of GC. And in 2018, Shu et al, identified a frequent proportion of CLDN18-ARHGAP fusion gene up to 25% in gastric signet ring cell carcinoma (SRCC) (Shu et al, 2018). It is also verified that the CLDN18-ARHGAP fusion gene is an oncogenic driver in GC, and patients with CLDN18-ARHGAP fusion were associated with worse survival outcomes and chemotherapy resistance (Shu et al, 2018). However, it remains unknown whether it could be a potential mutation target for neoantigen-based treatment.

The suppressive tumor immune microenvironment (TIME) has been considered a critical hindering factor for the satisfactory treatment response of immunotherapies among multiple solid tumors (Bagaev et al, 2021; Xiao et al, 2021). Recent studies have revealed that the oncogenic signaling pathways initiated or activated by driver gene alterations contribute largely to the formation of a suppressive TIME (Kalbasi et al, 2020; Peng et al, 2016; Spranger et al, 2015). For instance, the mutation, loss or amplification of driver genes like PIK3CA, PTEN, or AKT (Kalbasi et al, 2020; Peng et al, 2016; Zhang et al, 2017) can aberrantly activate PI3K-AKT-mTOR signaling and promote the accumulation and immunosuppressive function of regulatory T (Treg) cells and tumor-associated macrophages (TAMs) within the local tumor microenvironment to augment immune suppression (Isoyama et al, 2021; Kaneda et al, 2016; Sun et al, 2021). Nonetheless, it is still unclear whether the CLDN18-ARHGAP fusion gene could have immunosuppressive regulation on TIME as a driver gene alteration.

Here we report that CLDN18-ARHGAP fusions occur in 9% of Chinese GC patients and can generate immunogenic neoantigen peptides to induce NRT cells with robust and specific anti-tumor cytotoxicity. Also, we demonstrate that GC cells with CLDN18-ARHGAP fusion have increased free fatty acid (FFA) production caused by activation of PI3K-AKT-mTOR/FAS signaling, thus promoting the infiltration and survival of Treg cells within the TIME. Finally, based on the immunogenic and immunoregulatory properties of CLDN18-ARHGAP fusion, we propose that NRT cell infusion and/or PI3K inhibition as novel and promising immunotherapeutic interventions for GC.

# Results

## Recurrent CLDN18-ARHGAP gene fusions in Chinese GC patients

To evaluate the gene alternation landscape of Chinese GC, FFPE tumor tissues from 87 primary GC patients in our center were collected to perform NGS. TP53 (63%), ARID1A (23%), CDH1 (16%), APC (15%), LRP1B (13%), TGFBR2 (13%), FAT3 (11%), PIK3CA (9%), SMARCA4 (9%) and ACVR2A (8%) were the top 10 most frequently mutated SNV&INDEL genes, while CCNE1 (9%), FSR2 (8%) and MDM2 (8%) were the most frequent genes with CNVs in PC GC patients (Fig. 1A). It is worth noting that the CLDN18-ARHGAP gene fusion occurred in 9% (8/87) of GC patients and 19% (5/26) of poorly cohesive GC patients, which type with the worst prognosis. Four types of the CLDN18-ARHGAP fusion were identified in our study, including the breaks and links

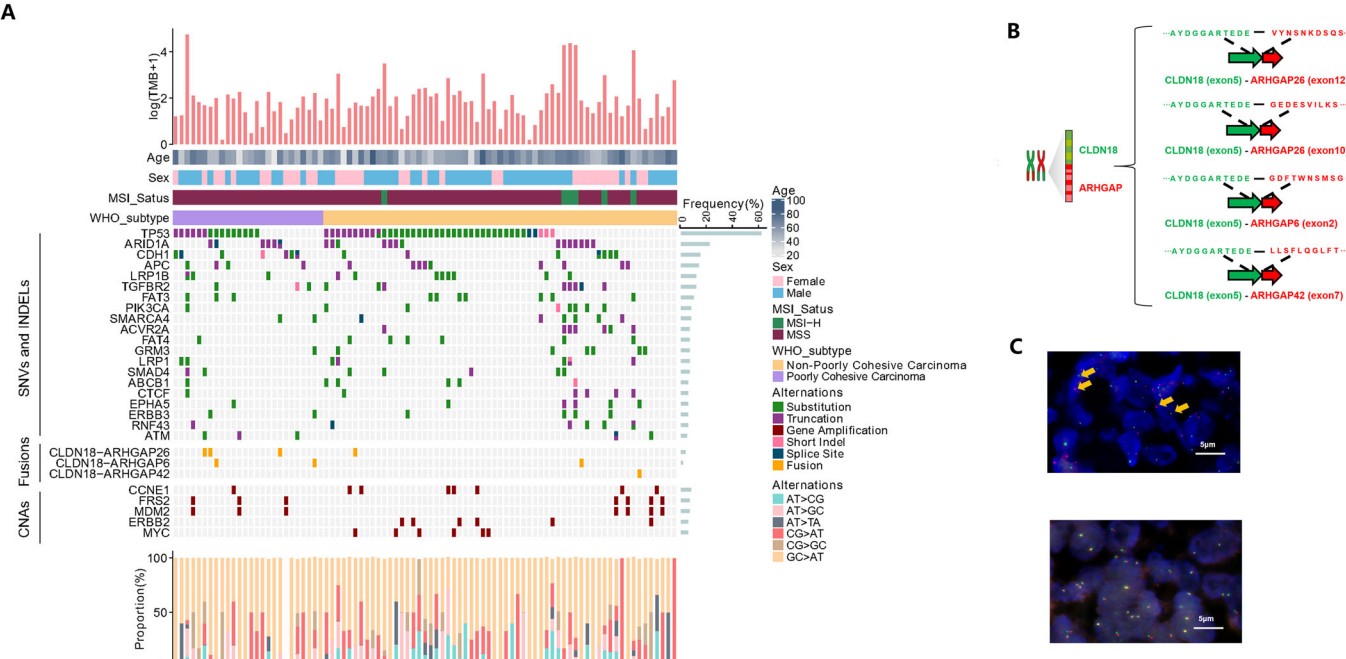

**Figure 1. Recurrent CLDN18-ARHGAP gene fusions in GC patients.**

(A) Tumor samples from 87 Chinese GC patients were collected for NGS, and the landscape of recurrent genetic alterations was shown. Patient information of age, sex, location, stage, MSI status and WHO type were provided on the top of the y axis, followed by the recurrent alterations, including SNV&INDELs, CNVs, and fusions. The frequency of each gene alteration was illustrated on the right of the mutation heat plot. (B) A diagram of CLDN18-ARHGAP fusion with amino acid sequences surrounding the junctions of different exons was shown. (C) FISH detection of the CLDN18 fusion. Red signals were located in exon 5 of the CLDN18 gene, and green signals were located after exon 5 of the CLDN18 gene, with yellow arrows indicating cells with breakings after exon 5 of the CLDN18 gene. Source data are available online for this figure.

of exon 5 from CLDN18, to exon 12 from ARHGAP26 ($N = 3$), exon 10 from ARHGAP26 ($N = 1$), exon 2 from ARHGAP6 ($N = 3$), or exon 7 from ARHGAP42 ($N = 1$) (Fig. 1A,B). Moreover, the presence of the CLDN18 fusion was also confirmed by us through fluorescence in situ hybridization (FISH) in tumor tissues of PC GC patients (Fig. 1C).

## CLDN18-ARHGAP gene fusion could derive an immunostimulatory peptide recognized by autologous and healthy donor's T cells

After identifying the increased occurrence of the CLDN18-ARHGAP fusion gene among GC patients, we next aimed to investigate whether neoantigens derived from CLDN18-ARHGAP fusions could elicit immunostimulatory T-cell responses. The NetMHCpan 4.0 was used to predict potential peptides derived from three CLDN18-ARHGAP fusion types previously reported to be present in tumor samples, including junctions of CLDN18 (exon 5)-ARHGAP26

(exon 12), CLDN18 (exon 5)-ARHGAP26 (exon 10) and CLDN18 (exon 5)-ARHGAP6 (exon 2) (Table 1; Appendix Tables S1 and S2). The candidate peptides were 9–11 amino acids long and predicted to bind to HLA-A*11:01, the most prevalent HLA-A type among Chinese populations. Corresponding wild-type (WT) peptides derived from CLDN18 and ARHGAP gene were used as controls (Table 1; Appendix Tables S1 and S2). First, PBMCs from patients with the corresponding CLDN18-ARHGAP fusion gene were collected to evaluate the immunogenicity of peptides derived from the fusion gene. Compared with WT and other candidate peptides, RTEDEVYNSNK (peptide 1, P1) derived from CLDN18 (exon 5)-ARHGAP26 (exon 12) fusion induced dramatically increased secretion of interferon-γ (IFN-γ) of more than twofold (Fig. 2A,B) and increased expression of CD137 on CD8$^+$ T cells from patient's autologous PBMCs (Fig. 2C). However, peptides derived from CLDN18 (exon 5)-ARHGAP26 (exon 10) or CLDN18 (exon 5)-ARHGAP6 (exon 2) fusions failed to trigger a significant increase of IFN-γ secretion from autologous PBMCs

**Table 1. Neoantigens derived from CLDN18/exon5–ARHGAP26/exon12.**

| Name | Gene | Sequence | Rank |
|------|------|----------|------|
| | CLDN18(e5)-ARHGAP26(e12) | YDGGARTEDEVYNSNKDSQS | |
| P1 | CLDN18(e5)-ARHGAP26(e12) | RTEDEVYNSNK | 1.7 |
| P2 | CLDN18(e5)-ARHGAP26(e12) | TEDEVYNSNK | 8.0 |
| P3 | CLDN18(e5)-ARHGAP26(e12) | RTEDEVYNS | 10.8 |
| P4 | CLDN18(e5)-ARHGAP26(e12) | GARTEDEVY | 13.8 |
| | CLDN18 | YDGGARTEDEVQSYPSKHDY | |
| WT1-1 | CLDN18 | RTEDEVQSYPS | 23.9 |
| WT2-1 | CLDN18 | TEDEVQSYPS | 58.3 |
| WT3-1 | CLDN18 | RTEDEVQSY | 1.3 |
| WT4-1 | CLDN18 | GARTEDEVQ | 45.0 |
| | ARHGAP26 | WMEAMDGREPVYNSNKDSQS | |
| WT1-2 | ARHGAP26 | DGREPVYNSNK | 11.8 |
| WT2-2 | ARHGAP26 | GREPVYNSNK | 10.9 |
| WT3-2 | ARHGAP26 | DGREPVYNS | 51.3 |
| WT4-2 | ARHGAP26 | AMDGREPVY | 3.2 |

Amino acid from CLDN18 and ARHGAP26 gene were indicated with green and red letters, respectively. P1, P2, P3, and P4 were versicolor sequences, representing neoantigens from CLDN18/exon5–ARHGAP26/exon12; WT1-1, WT2-1, WT3-1, and WT4-1 were green sequences, representing wild-type peptides from CLDN18 gene; WT1-2, WT2-2, WT3-2, and WT4-2 were red sequences, representing wild-type peptides from ARHGAP26 gene.

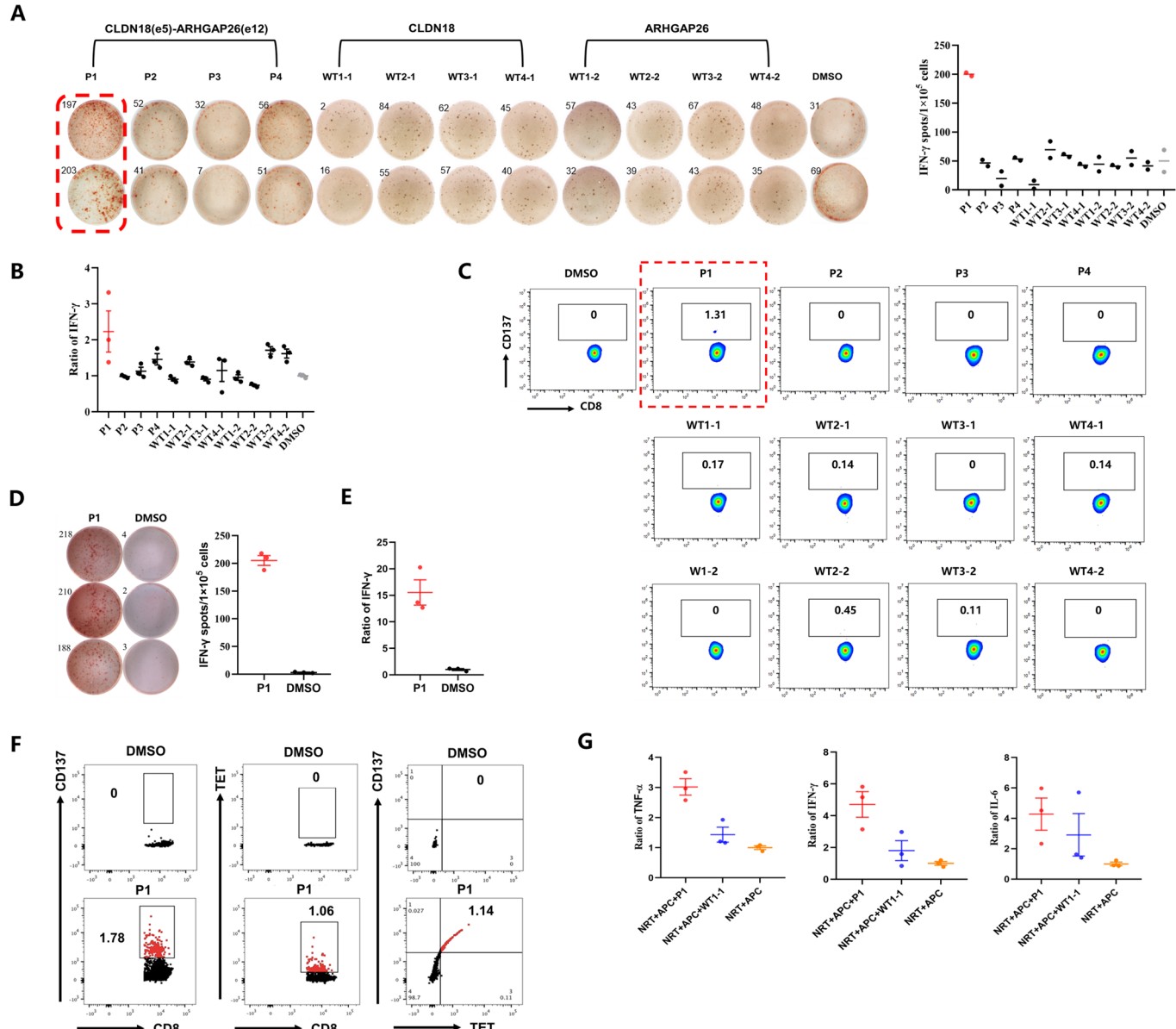

**Figure 2. CLDN18-ARHGAP gene fusion generates an immunostimulatory peptide recognized by autologous and donors' T cells.**

Autologous PBMCs from one PC GC patient with HLA-A*11:01 typing and CLDN18(e5)-ARHGAP26(e12) fusion gene were isolated and added to 3 separate experiment holes. They were stimulated with CLDN18(e5)-ARHGAP26(e12)-derived peptides, CLDN18-derived WT peptides or ARHGAP26-derived WT peptides every 3 days in the presence of IL-2, and T-cell response to each peptide were analyzed on day 10 by (A) IFN-γ ELISPOT assays and quantification of IFN-γ spots ($n = 2$ biological replicates), (B) quantification of IFN-γ in CBA assays ($n = 3$ biological replicates), and (C) detection of CD137 expression on CD8$^+$ T cells via flow cytometry. T cells from one healthy donor were isolated and stimulated with APCs loading P1 every 7 days in the presence of IL-2. After 2-round stimulation, T-cell response to P1 was analyzed on day 15 by (D) IFN-γ ELISPOT assays and the quantification of IFN-γ spots ($n = 3$ biological replicates), (E) quantification of IFN-γ in CBA assays ($n = 3$ biological replicates), and (F) detection of CD137 and P1-specific tetramer expression on CD8$^+$ T cells via flow cytometry. (G) Immunostimulatory cytokines including TNF-α, IFN-γ, and IL-6 from the culture of donor T cells stimulated with IL-2 and APCs loading P1 or WT peptides were detected by CBA. The increase of IFN-γ, TNF-α, or IL-6 more than twofold was considered different ($n = 3$ biological replicates). Data information: The data with error bars are shown as mean ± SEM. Source data are available online for this figure.

more than two times or induce higher proportions of CD137$^+$ CD8$^+$ T cells (Appendix Figs. S1A–D and S2A–D).

To avoid the impacts of chemotherapy on PBMCs from GC patients, we further explored whether P1 could lead to the induction of NRT cells derived from healthy donors that specifically target the CLDN18 (exon 5)-ARHGAP26 (exon 12) fusion. DCs isolated from healthy donors with HLA-A*11:01

typing were loaded with or without peptides and then cocultured with donor T cells. The results demonstrated that after two cycles of induction and stimulation, IFN-γ levels in cell culture with P1 increased by more than 60-fold by ELISPOT detection and more than 15-fold by CBA measurement compared with the DMSO group (Fig. 2D,E). In addition, prominent upregulation of CD137 and P1-specific tetramer expression was also observed and highly

overlapped in donor T cells stimulated with P1 (Fig. 2F). Moreover, compared with wild-type peptide (WT1-1) or non-peptide groups, the secretions of pro-inflammatory cytokines including TNF-α, IFN-γ, and IL-6 were significantly increased in T cells with P1 stimulation (Fig. 2G). However, there was no significant change in the levels of anti-inflammatory cytokines such as IL-4, IL-5, IL-10, and IL-13 (Appendix Fig. S3). Moreover, we also found that peptides derived from CLDN18 (exon 5)-ARHGAP26 (exon 10) or CLDN18 (exon 5)-ARHGAP6 (exon 2) fusions could not effectively induce activation of the healthy donors' T cells (Appendix Figs. S1E–G and S2E–G). Together, these results strongly indicate the potential immunogenicity of P1 peptide to generate CLDN18 (exon 5)-ARHGAP26 (exon 12) fusion-specific NRTs with immunostimulatory responses.

## NRT cells targeting CLDN18-ARHGAP fusion exhibit specific anti-tumor cytotoxicity

The effective induction of immunostimulatory T-cell responses by P1 prompted us to further investigate whether this immunogenic peptide could also enhance the specific tumor-killing ability of NRT cells. By using the pLVX-puro transfection system, we successfully triggered the expression of CLDN18 (exon 5)-ARHGAP26 (exon 12) fusion gene in human GC cell line SNU601 which naturally harbors the HLA-A*11:01 typing (SNU601-OE), and both fusion gene and HLA-A*11:01 expression in another human GC cell line MKN45 (MKN45-OE), which are both confirmed by FISH detection (Fig. 3A). We found in vitro that compared with non-peptide-stimulated T cells, P1-induced NRT cells exhibited remarkably enhanced ability to induce SNU601-OE ($P = 0.0008$) and MKN45-OE ($P = 0.0057$) cell apoptosis at multiple E:T ratios (10:1) (Fig. 3B,C).

To further validate the anti-tumor function of P1-induced NRT cells, the GC xenograft mouse model implanted with MKN45-OE cells was established by us, followed by i.v. infusion of T or NRT cells induced and expanded in vitro (Fig. 3D). SNU601-OE cells were challenging to grow subcutaneously in nude mice and therefore not selected. Consistent with the results of cytotoxic assays, the growth of MKN45-OE tumors was significantly reduced after NRT cell adoptive treatment (Fig. 3E,F), with good biosafety and no apparent toxicity during the treatment period (Fig. 3G,H). We also constructed the MKN45 transfected with control-plasmid (MKN45-NC)-challenged xenograft model and observed that infusion of P1-induced NRT cells could not lead to increased tumor reduction compared with non-peptide-stimulated T cells (Appendix Fig. S4). Collectively, both the in vitro and in vivo results demonstrate that the immunogenic P1 peptide derived from CLDN18 (exon 5)-ARHGAP26 (exon 12) fusion can generate NRT cells with specific and promising cytotoxicity against target tumor cells.

## CLDN18-ARHGAP gene fusion contributes to a suppressive TIME by upregulating Treg cells

Apart from the potential of CLDN18-ARHGAP fusion to generate immunogenic peptides and NRTs, we next investigated whether CLDN18-ARHGAP fusion could have direct impacts on the TIME of GC. The mouse forestomach carcinoma (MFC) cells were transduced with CLDN18-ARHGAP fusion (MFC-OE) or control

plasmids (MFC NC) by using the pLVX-puro system, and the immunocompetent GC mouse model was established by s.c. challenge of MFC-OE or MFC NC in 615-line mice. Upon termination 2 weeks later, we analyzed the immune competence collected in tumors, spleens, and draining lymphocyte nodes from the MFC-OE or MFC-NC group. Flow cytometry analysis showed that compared to the MFC-NC group, the MFC-OE group had significantly increased infiltrations of Tregs within both tumors and draining lymphocyte nodes (Fig. 4A,C). Meanwhile, there was no significant difference in other immune cells' intratumoral or splenic infiltrations, including CD8+ T cells, macrophages, MDSCs, or DCs (Fig. 4A,B).

## CLDN18-ARHGAP fusion promotes Treg cell survival through increased FFA production via PI3K/AKT-mTOR-FAS signaling activation in GC cells

Multiple pieces of research have revealed that alternations of driver genes could lead to the metabolism reprogramming of cancer cells to promote immune escape and tumor progression (Kalbasi et al, 2020; Peng et al, 2016; Spranger et al, 2015). Since FFAs are documented to be critical metabolites for Treg cells that favor their survival and proliferation (Muroski et al, 2017; Yan et al, 2022). we next tried to explore whether GC cells with CLDN18-ARHGAP fusion facilitate Treg infiltration in the TIME in a metabolism-dependent manner. We first detected the FFA concentrations in the culture supernatants of GC cell lines (MKN45, SNU601, and MFC) and tumor tissues of MFC cell lines with or without CLDN18-ARHGAP fusion (MKN45/SNU601/MFC NC/OE). We found significantly increased production of FFAs among MKN45-OE, SNU601-OE, and MFC-OE cells and tumor tissues than their NC counterparts (Fig. 5A,B). After the coculture of MKN45-OE cells with PBMCs with the addition of fatty acid scavenger (FAI), an evident decrease of FFA concentration was detected in the coculture supernatants (Fig. 5C). Moreover, FAI administration decreased the proportion of Treg (FOXP3+CD25+/CD4+) cells among PBMCs and enhanced the ability of T cells derived from PBMCs to induce tumor cell apoptosis (Fig. 5D,E), indicating that FFA production from MFC-OE cells is essential for the survival and immunosuppressive functions of Treg cells.

To further explore the internal mechanisms of the increased FFA production in GC cells with CLDN18-ARHGAP fusion, the key enzyme of de novo fatty acid synthesis, FAS, and the key components of its upstream regulatory signaling pathway PI3K/AKT-mTOR-FAS were investigated (Kumagai et al, 2020). Results of western blot analysis showed that compared to MKN45/SNU601-NC cells, the expressions of p-AKT, p-mTOR, Raptor, SREBP-1, and FAS were remarkably upregulated in MKN45/SNU601-OE cells (Fig. 5F), suggesting the activation of PI3K/AKT-mTOR-FAS signaling. Next, we found that a PI3K inhibitor (PI3Ki)—Pictilisib could reverse PI3K/AKT-mTOR-FAS signaling activation (Fig. 5F), and directly inhibit the proliferation of GC cells with CLDN18-ARHGAP fusion (Appendix Fig. S5). Also, the administration of PI3Ki in NRT cells significantly decreased FFA concentration and the proportion of Treg cells (Fig. 5G,H). Moreover, when adding PI3Ki into the coculture of MKN45-OE cells with NRTs, the killing capacity of NRTs against targeted tumor cells was improved, which could be reversed by the extra FFA supplementation (Fig. 5I). Taken together, these results

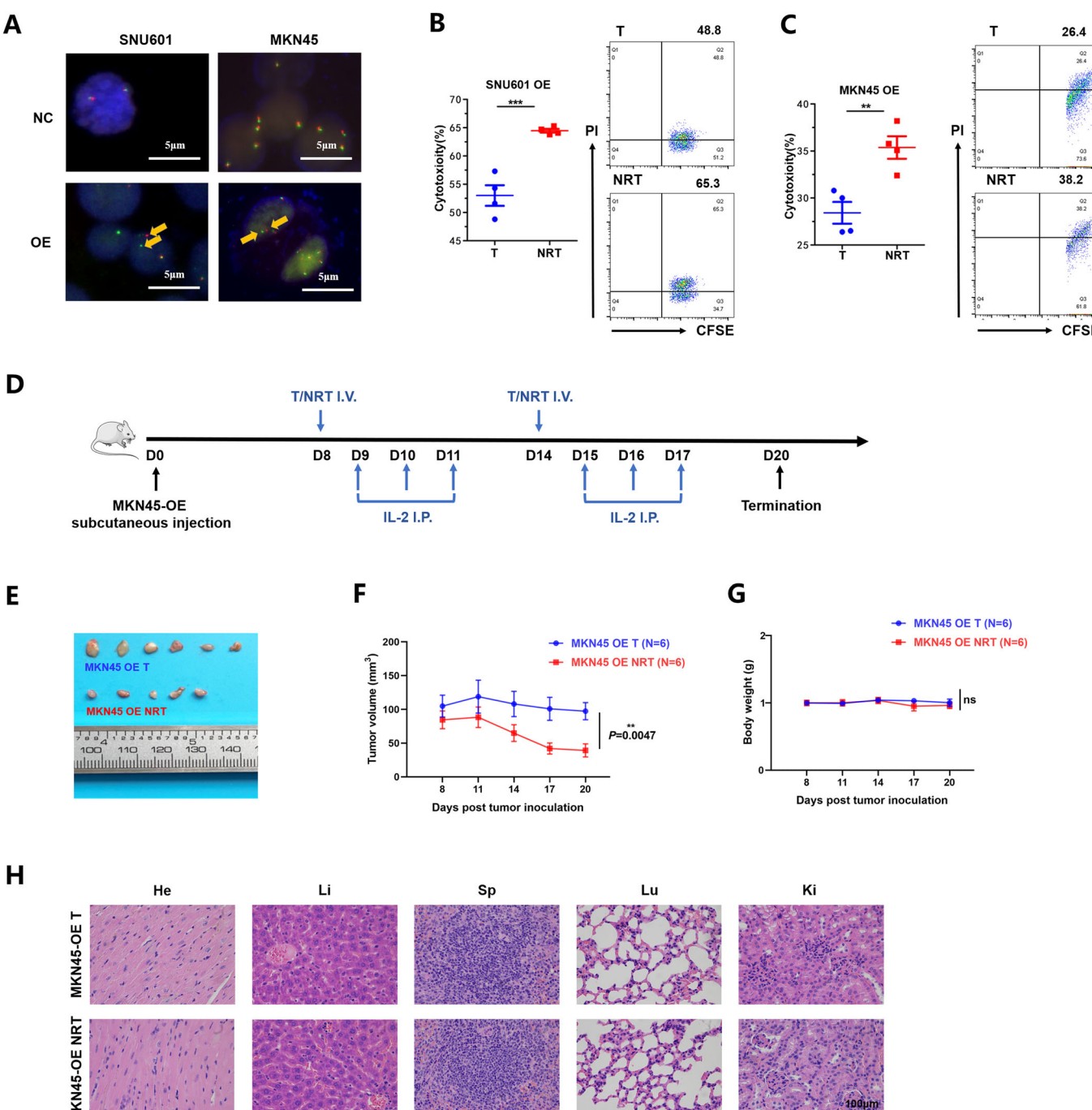

**Figure 3.  NRT cells induced by the immunogenic peptide derived from CLDN18-ARHGAP fusion exhibit specific anti-tumor cytotoxicity.**

(A) FISH detection of the CLDN18 fusion in SNU601-NC/OE and MKN45-NC/OE cells. Red signals were located in exon 5 of the CLDN18 gene, and green signals were located after exon 5 of the CLDN18 gene, with yellow arrows indicating cells with breakings after exon 5 of the CLDN18 gene. Lysis of monolayer CSFE-labeled SNU601-OE ($P = 0.0008$) (B) and MKN45-OE ($P = 0.0057$) (C) cells cocultured with NRT or T cells at an E:T ratio of 10:1 were detected by flow cytometry ($n = 4$ biological replicates). (D) Schematic of NRT cell treatment schedule in MKN45-OE xenograft model. Balb/c nude mice ($n = 6$ per group) were injected s.c. with $10^7$ MKN45-OE cells and treated i.v. with T or NRT cells twice every 6 days. (E) Mice were sacrificed at the treatment endpoint and tumors were removed for analysis. One tumor sample removed from MKN45-OE-challenged Balb/c mice treated with NRT cells was too small to be dissected. (F) Tumor volumes of MKN45-OE-challenged Balb/c mice treated with T or NRT cells ($n = 6$ biological replicates, $P = 0.0047$). (G) Body weight of mice in MKN45-OE xenograft model ($n = 6$ biological replicates, $P = 0.5837$). (H) Safety evaluation of T or NRT cell treatment in mouse organs was shown by hematoxylin and eosin (H&E) staining (Scale bars, 100 μm). Data information: The data with error bars are shown as mean ± SEM. ns not significantly, **$P < 0.01$, ***$P < 0.001$ by two-tailed unpaired-sample Student $t$ test. Source data are available online for this figure.

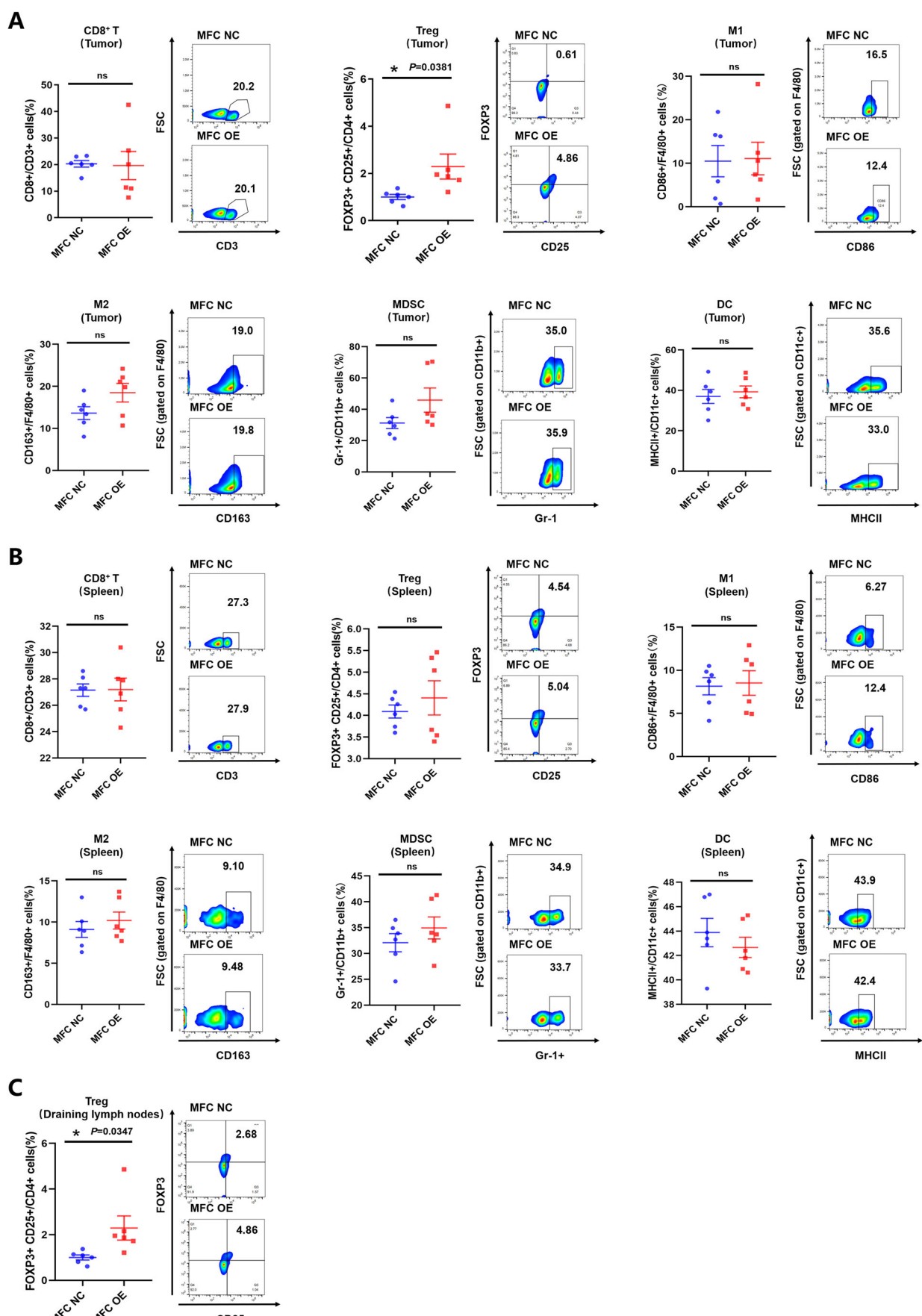

◀ **Figure 4. CLDN18-ARHGAP fusion contributes to the formation of a suppressive tumor immune microenvironment by upregulating Treg cells.**

615-line mice were injected s.c. with $10^6$ MFC NC/OE cells ($n = 6$ per group). Mice were sacrificed at the treatment endpoint and tumors, spleens and draining lymph nodes were removed for analysis. (A) The proportions of CD8$^+$/CD3$^+$ cells ($P = 0.9077$), FOXP3$^+$ CD25$^+$/CD4$^+$ cells ($P = 0.0381$), CD86$^+$/F4/80$^+$ cells ($P = 0.9108$), CD163$^+$/F4/80$^+$ cells ($P = 0.1002$), Gr-1$^+$/CD11b$^+$ cells ($P = 0.1135$) and MHC-II$^+$/CD11c$^+$ cells ($P = 0.6299$) in tumors ($n = 6$ biological replicates) from MFC NC or MFC OE group were determined by flow cytometry. (B) The proportions of CD8$^+$/CD3$^+$ cells ($P = 0.9603$), FOXP3$^+$ CD25$^+$/CD4$^+$ cells ($P = 0.4730$), CD86$^+$/F4/80$^+$ cells ($P = 0.8332$), CD163$^+$/F4/80$^+$ cells ($P = 0.4574$), Gr-1$^+$/CD11b$^+$ cells ($P = 0.3215$) and MHC-II$^+$/CD11c$^+$ cells ($P = 0.4136$) in spleens ($n = 6$ biological replicates) from MFC NC or MFC OE group were determined by flow cytometry. (C) The proportions of FOXP3$^+$ CD25$^+$/CD4$^+$ cells in draining lymph nodes from MFC NC or MFC OE group were determined by flow cytometry ($P = 0.0347$, $n = 6$ biological replicates). Data information: The data with error bars are shown as mean ± SEM. ns, not significantly, *$P < 0.05$ by two-tailed unpaired-sample Student $t$ test. Source data are available online for this figure.

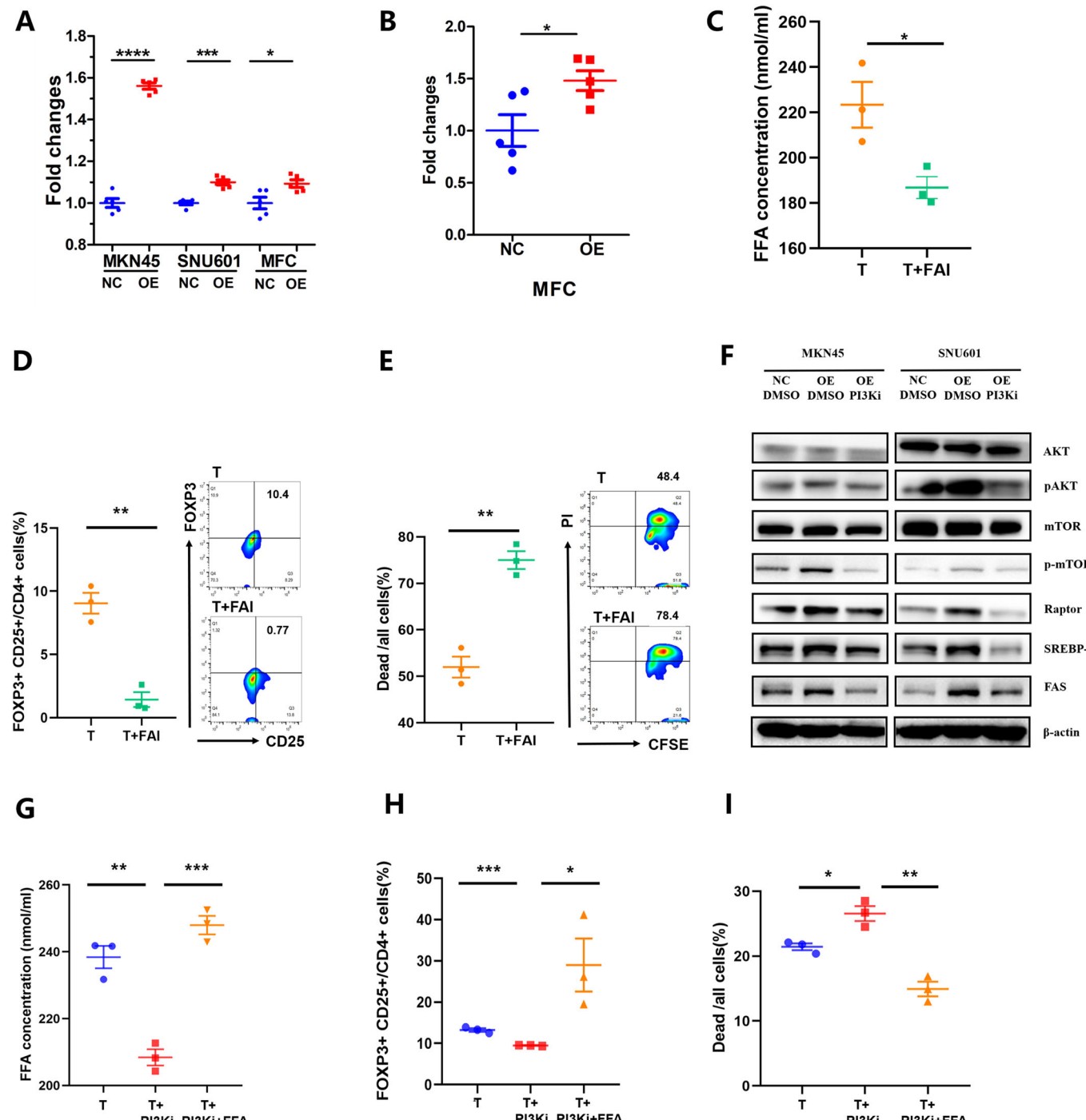

**Figure 5. CLDN18-ARHGAP fusion promotes Treg cell survival through increased FFA production via PI3K/AKT-mTOR-FAS signaling activation in GC cells.**

(A) FFA concentrations in the culture supernatants of SNU601-NC/OE ($P < 0.0001$), MKN45-NC/OE ($P = 0.0002$) and MFC NC/OE ($P = 0.0237$) were detected ($n = 5$ biological replicates). (B) MFC-NC/OE cells ($1 \times 10^6$) were injected s.c. into 615-line mice, and tumor tissues were extracted on day 15. FFA concentrations in MFC tumors were detected ($P = 0.0286$, $n = 5$ biological replicates). Free fatty scavenger (FFI) was added into the coculture of MKN45-OE and PBMCs, and then (C) FFA concentrations in the coculture ($P = 0.0305$, $n = 3$ biological replicates), (D) the proportions of FOXP3[+] CD25[+]/CD4[+] cells ($P = 0.0016$, $n = 3$ biological replicates) and (E) lysis of monolayer CSFE-labeled MKN45-OE cells were detected ($P = 0.0014$, $n = 3$ biological replicates). (F) Western blot analysis of PI3K/AKT-mTOR-FAS signaling components including AKT, p-AKT, mTOR, p-mTOR, Raptor, SREBP-1 and FAS in MKN45/SNU601-NC, MKN45/SNU601-OE or MKN45/SNU601-OE treated with PI3Ki. PI3Ki and/or FFAs were added into the coculture of MKN45-OE and T cells and then (G) FFA concentrations in the coculture (T vs T + PI3Ki, $P = 0.0020$; T + PI3Ki vs T + PI3Ki+FFA, $P = 0.0005$; $n = 3$ biological replicates), (H) the proportions of FOXP3[+] CD25[+]/CD4[+] cells (T vs T + PI3Ki, $P = 0.0008$; T + PI3Ki vs T + PI3Ki+FFA, $P = 0.0386$; $n = 3$ biological replicates) and (I) lysis of monolayer CSFE-labeled MKN45-OE cells were detected (T vs T + PI3Ki, $P = 0.0156$; T + PI3Ki vs T + PI3Ki+FFA, $P = 0.0020$; $n = 3$ biological replicates). Data information: The data with error bars are shown as mean ± SEM. *$P < 0.05$, **$P < 0.01$, ***$P < 0.001$, ****$P < 0.0001$ by two-tailed unpaired-sample Student $t$ test. Source data are available online for this figure.

suggest that GC cells with CLDN18-ARHGAP fusion promotes Treg cell survival and weakens anti-tumor cytotoxicity of NRTs through increased FFA production caused by the PI3K/AKT-mTOR-FAS signaling activation.

## PI3K inhibition triggers anti-tumor effects and improves the tumor immune microenvironment in the GC model with CLDN18-ARHGAP fusion

Given that the CLDN18-ARHGAP fusion gene activates the PI3K/AKT-mTOR-FAS signaling to promote Treg cell survival in vitro, we next further explored the effects of targeting PI3K signaling in vivo. In the syngeneic MFC OE/NC-challenged GC subcutaneous or abdominal model, the PI3K inhibitor Pictilisib (75 mg/kg) or DMSO was i.p. administrated every 2 days for 2 weeks (Fig. 6A).

For the subcutaneous model, the body weights of mice receiving various treatments were not significantly different (Fig. 6B). We found that PI3Ki treatment resulted in obviously reduced tumor growth in MFC OE-challenged mice ($P = 0.0482$), but showed no significant difference in MFC-NC-challenged mice (Fig. 6C,D), indicating the specific targeting and function of PI3Ki on MFC OE tumors. As to the TIME analysis, we observed that compared to the MFC-NC group, the proportions of Treg cells in tumors and draining lymphocyte nodes were both increased in the MFC OE group, while PI3Ki administration effectively reversed the upregulated Treg infiltration in both draining lymphocyte nodes and tumor tissues from the MFC OE group ($P = 0.0011$ draining LN, $P = 0.0223$ primary, Fig. 6E,F). In addition, PI3Ki treatment also increased the intratumoral infiltration of CD8[+] T cells in the MFC OE group and decreased M1-like macrophage proportion in the spleens of the MFC-NC group. At the same time, there was no significant change of other immune cells, including M2 macrophage, DCs and MDSCs in tumor or spleen tissues among the four groups (Fig. 6F; Appendix Fig. S6A).

Furthermore, we explored the anti-tumor effects of targeting PI3K signaling in the syngeneic MFC OE/NC-challenged GC abdominal model (Fig. 6A). It demonstrated that PI3Ki treatment could obviously reduce the tumor growth in MFC OE-challenged mice ($P = 0.0275$) but had no significant impact on MFC-NC-challenged mice (Fig. 6G,H). Moreover, it showed that Tregs in the TIME and tumor-draining lymph nodes were significantly upregulated in MFC-OE compared to MFC NC ($P < 0.0001$ primary, $P = 0.0160$ draining LN, Fig. 6J,I). And PI3Ki could downregulate Tregs in both the TIME and tumor-draining lymph nodes only in the MFC-OE group ($P = 0.0151$ primary, $P = 0.0469$ draining LN, Fig. 6J,I). In addition, PI3Ki treatment also elevated

the proportion of M1-like macrophages and downregulated the proportion of DCs in the spleens of the MFC-NC group (Appendix Fig. S6B).

Therefore, we also confirmed in vivo that inhibiting the PI3K/AKT-mTOR-FAS signaling could specifically trigger anti-tumor effects and downregulate Treg infiltration in the TIME of the GC model with CLDN18-ARHGAP fusion.

## Discussion

TCGA describes a comprehensive molecular evaluation of 295 primary gastric adenocarcinomas. The top mutated genes are TP53 (50%), ARID1A (14%), PIK3CA (12%), CDH1 (11%) and SMAD4 (8%) (Bass et al, 2014). Also, the GC molecular subtype of the Asian Cancer Research Group (ACRG) was analyzed based on NGS results of 300 primary GC tumor specimens. TP53 (33%), ATN1 (25%), MUC6 (22%), ARID1A (18%), and PIK3CA (14%) were the most significantly mutated genes (Cristescu et al, 2015). In our cohort, TP53 (63%), ARID1A (23%), CDH1 (16%), APC (15%), LRP1B (13%), TGFBR2 (13%), FAT3 (11%), PIK3CA (9%), SMARCA4 (9%) and ACVR2A (8%) were the top 10 most frequently mutated SNV&INDEL genes, while CCNE1 (9%), FSR2 (8%) and MDM2 (8%) were the most frequent genes with CNVs in PC GC patients. We can see that TP53, ARID1A, and PIK3CA were the most commonly mutated genes in different populations.

The promising results of several clinical trials have led to the approval of checkpoint inhibitors for treating specific GC patients in front- to back-line treatments (Joshi et al, 2021). Though biomarkers, such as MSI-H/dMMR, PD-L1, and tumor mutation burden, can be used for stratifying potentially responding candidates, checkpoint inhibitors only derive clinical benefits in a limited subset of GC patients. New therapeutic strategies have emerged, as a recent clinical report on CAR-T adoptive therapy targeting CLDN18.2 showed an objective response rate of 48.6% for CLDN18.2 positive GC patients (Qi et al, 2022). However, no studies focus on recurrent oncogenic CLDN18-ARHGAP fusion gene in GC.

Gene fusions refer to chimeric genes connecting the coding region of two or more genes end to end under the same regulatory sequence, which plays a crucial role in tumor diagnosis (Gao et al, 2018) and prognosis prediction (Kalina et al, 2017; Tomlins et al, 2011). Only limited fusion genes in specific cancer types can be pharmaceutically targeted currently (Dai et al, 2018; Druker, 2008; Mertens et al, 2015; Rutkowski et al, 2010). With the development of NGS techniques, fusion genes have manifested their importance in generating tumor neoantigens. Since peptides spanning the

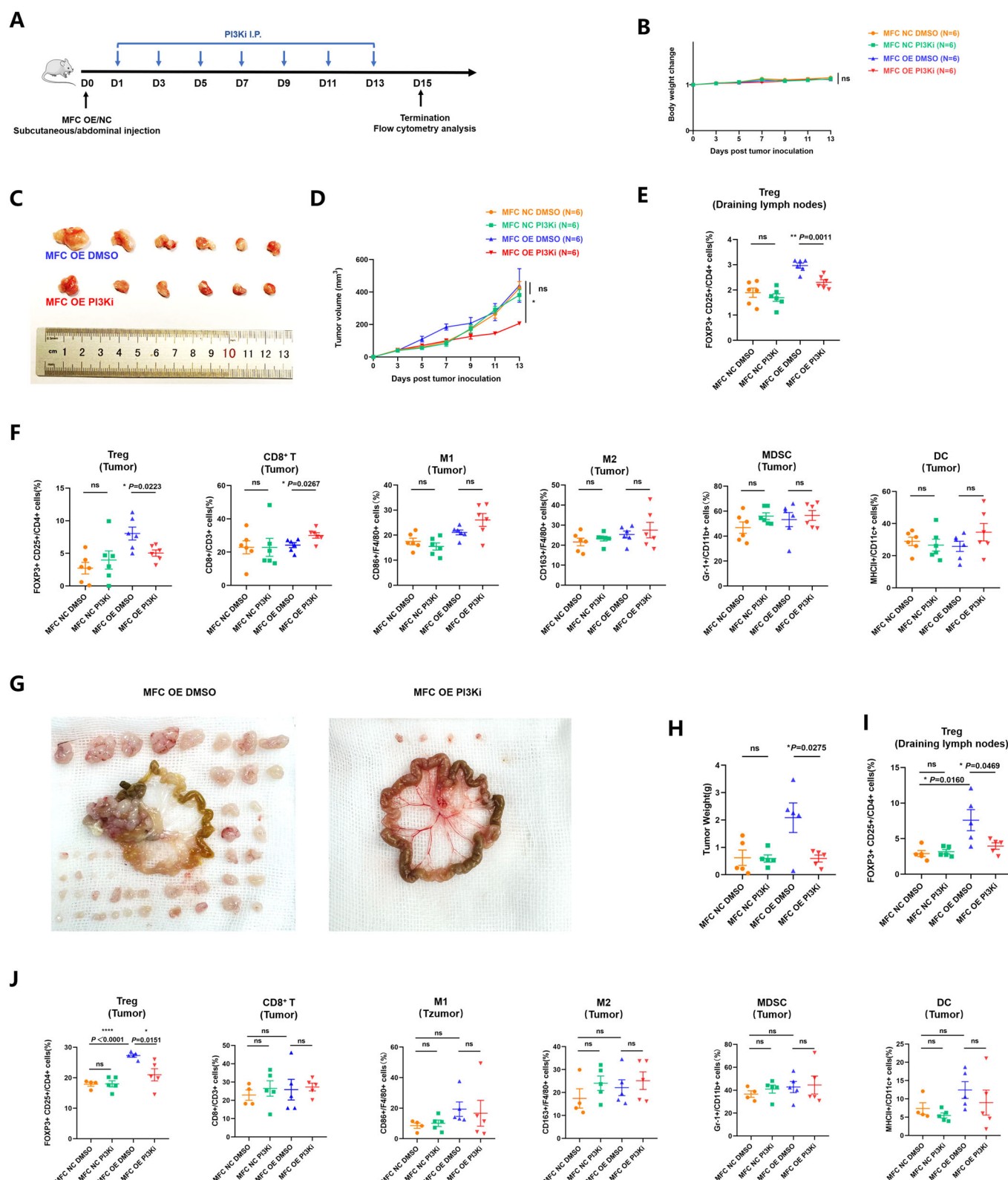

breakpoint regions of two genes are vastly dissimilar from self-antigens, gene fusions can theoretically serve as ideal sources of neoantigens. Recent studies have started to explore the feasibility of fusion gene-derived neoantigen therapeutics. In 2018, Jennifer et al,

(Kalina et al, 2017) verified that TMPRSS: ERG type VI (2:4) fusion peptides could expand specific T cells from healthy HLA-A*02:01 donors. Later, Yang et al, (Yang et al, 2019) identified that neoantigens derived from MYB-NFIB fusion genes could induce

**Figure 6. PI3K inhibition triggers anti-tumor effects and improves tumor immune microenvironment in the GC model with CLDN18-ARHGAP fusion.**

(A) Schematic of PI3Ki treatment schedule in MFC OE/NC-challenged syngeneic GC model. 615-line mice are s.c. or i.p. injected with MFC OE/NC cells ($n$ = 6 or 5 per group) and treated i.p. with PI3Ki (75 mg/kg) or DMSO every other day for 2 weeks. Mice were sacrificed at the treatment endpoint, tumors and draining lymph nodes were removed for analysis. One tumor sample removed from MFC NC abdominally challenged 615-line mice treated with DMSO was too small to prepare cell suspension. (B) Body weight of MFC OE/NC subcutaneously challenged 615-line mice treated with PI3Ki or DMSO (MFC NC $P$ = 0.2458, MFC OE $P$ = 0.9023, $n$ = 6 per group). (C) Tumors removed from MFC OE subcutaneously challenged 615-line mice treated with PI3Ki or DMSO. (D) Tumor volumes of MFC OE/NC subcutaneously challenged 615-line mice treated with PI3Ki or DMSO (MFC NC $P$ = 0.4513, MFC OE $P$ = 0.0482, $n$ = 6 per group). (E) The proportions of FOXP3$^+$ CD25$^+$/CD4$^+$ cells in draining lymph nodes of MFC OE/NC subcutaneous challenged 615-line mice were determined by flow cytometry (MFC NC $P$ = 0.4297, MFC OE $P$ = 0.0011, $n$ = 6 biological replicates). (F) The proportions of FOXP3$^+$ CD25$^+$/CD4$^+$ cells (MFC NC $P$ = 0.4663, MFC OE $P$ = 0.0223), CD8$^+$/CD3$^+$ cells (MFC NC $P$ = 0.9949, MFC OE $P$ = 0.0267), CD86$^+$/F4/80$^+$ cells (MFC NC $P$ = 0.3249, MFC OE $P$ = 0.1023), CD163$^+$/ F4/80$^+$ cells (MFC NC $P$ = 0.4670, MFC OE $P$ = 0.6170), Gr-1$^+$/CD11b$^+$ cells (MFC NC $P$ = 0.1147, MFC OE $P$ = 0.6427) and MHC-II$^+$/CD11c$^+$ cells (MFC NC $P$ = 0.6157, MFC OE $P$ = 0.1919) in tumors of MFC OE/NC subcutaneous challenged 615-line mice were determined by flow cytometry ($n$ = 6 biological replicates). (G) Tumors removed from MFC OE abdominally challenged 615-line mice treated with PI3Ki or DMSO. (H) Tumor weights of MFC OE/NC abdominally challenged 615-line mice treated with PI3Ki or DMSO (MFC NC $P$ = 0.9281, MFC OE $P$ = 0.0275, $n$ = 5 per group). (I) The proportions of FOXP3$^+$ CD25$^+$/CD4$^+$ cells in draining lymph nodes of MFC OE/NC abdominally challenged 615-line mice were determined by flow cytometry (MFC NC DMSO vs MFC-NC PI3Ki, $P$ = 0.6346; MFC OE DMSO vs MFC OE PI3Ki, $P$ = 0.0469; MFC-NC DMSO vs MFC OE DMSO, $P$ = 0.0160; $n$ = 5 biological replicates). (J) The proportions of FOXP3$^+$ CD25$^+$/CD4$^+$ cells (MFC-NC DMSO vs MFC-NC PI3Ki, $P$ = 0.9406; MFC OE DMSO vs MFC OE PI3Ki, $P$ = 0.0151; MFC-NC DMSO vs MFC OE DMSO, $P$ <0.0001), CD8$^+$/CD3$^+$ cells (MFC-NC DMSO vs MFC-NC PI3Ki, $P$ = 0.5234; MFC OE DMSO vs MFC OE PI3Ki, $P$ = 0.8194; MFC-NC DMSO vs MFC OE DMSO, $P$ = 0.6781), CD86$^+$/F4/80$^+$ cells (MFC-NC DMSO vs MFC-NC PI3Ki, $P$ = 0.5976; MFC OE DMSO vs MFC OE PI3Ki, $P$ = 0.7889; MFC-NC DMSO vs MFC OE DMSO, $P$ = 0.0933), CD163$^+$/F4/80$^+$ cells (MFC-NC DMSO vs MFC-NC PI3Ki, $P$ = 0.2401; MFC OE DMSO vs MFC OE PI3Ki, $P$ = 0.5715; MFC-NC DMSO vs MFC OE DMSO, $P$ = 0.4163), Gr-1$^+$/CD11b$^+$ cells (MFC-NC DMSO vs MFC-NC PI3Ki, $P$ = 0.2743; MFC OE DMSO vs MFC OE PI3Ki, $P$ = 0.4268; MFC-NC DMSO vs MFC OE DMSO, $P$ = 0.1301) and MHC-II$^+$/CD11c$^+$ cells in tumors of MFC OE/NC abdominally challenged 615-line mice were determined by flow cytometry ($n$ = 5 biological replicates). Data information: The data with error bars are shown as mean ± SEM. ns, not significantly, *$P$ < 0.05, **$P$ < 0.01, ****$P$ < 0.0001 by two-tailed unpaired-sample Student $t$ test. Source data are available online for this figure.

cytotoxic T-cell responses. Furthermore, DEK-AFF2 fusion, identified in head and neck cancer, was found to prime fusion-derived-neoantigen-specific cytotoxicity and revive efficacies of checkpoint inhibitors in tumors with low mutation burden and barren immune infiltration (Yang et al, 2019). In our study, we identified an HLA-A*11:01-binding immunogenic neoantigen peptide P1 (RTEDEVYNSNK) derived from CLDN18-ARHGAP gene fusion and successfully induced NRT cells with strong tumor-killing ability in both cell lines and mouse models with CLDN18-ARHGAP gene fusion. Thus, our study fills an essential bit of the overall puzzle that fusion-associated neoantigen is a significant source of tumor-specific targets that can elicit immune reactions in GC. We also display a highly hopeful prospect of applications in adoptive T-cell therapy, tumor vaccines, or other immunotherapies in clinic, which extend personalized treatment options for GC patients with CLDN18-ARHGAP gene fusion.

Immunosuppressive tumor microenvironments are reported to impede the efficacies of immunotherapies such as checkpoint inhibitors, neoantigen vaccines and NRT adoptive therapy (Galon et al, 2019). Accumulating evidence have revealed that gene fusions may interact mutually with the tumor immune milieu and mediate tumor immune escape (Ota et al, 2015; Wang et al, 2021) by hampering cytolytic immune functions and infiltration (Kalina et al, 2017; Yang et al, 2019). Among all immune components in the TME, Treg cells are a crucial class of immunosuppressive cells, which inhibit cytotoxic immune cells by various mechanisms (Wing et al, 2019). Antigen stimulation and selection can also clonally expand neoantigen-specific Treg cells in the TIME, hampering immune responses induced by neoantigen vaccines (Blass et al, 2021). Here, we demonstrated that the CLDN18-ARHGAP fusion gene contributed to an immunosuppressive TIME by enhancing the accumulation of Treg cells in the TIME, which may impede CLDN18-ARHGAP-derived neoantigen therapeutic effects.

Immune cells in the TME have distinguished metabolic phenotypes to maintain their survival and functions (Xia et al, 2021). Treg cells harbor a special metabolic mode, taking in FFA more eminently than other T-cell subsets and highly engaging in

fatty acid oxidation and oxidative phosphorylation for energy supply (Muroski et al, 2017; Yan et al, 2022). In comparison, activated effector T cells upregulate glycolysis to support cell growth and relatively lower fatty acid oxidation (FAO) levels (Xia et al, 2021). Metabolic reprogramming is also a hallmark of cancer (Pavlova et al, 2022) and can regulate immune cell constitutions and functions by shifting the concentration and species of metabolites to tune a more immunosuppressive microenvironment (Wagner et al, 2019). Abundant FFAs produced by tumor cells can provide a metabolic advantage for Treg cells survival (Kumagai et al, 2020) while blocking FFA uptake of Treg cells by ablating CD36, a scavenger molecule on the cell membrane, can attenuate their suppressive functions (Wang et al, 2020). Our results elucidated that GC with CLDN18-ARHGAP fusion can enhance immune suppression by increasing FFA production, undermining T-cell cytotoxicity, and promoting the survival of Treg cells.

Cancer metabolic reprogramming is closely linked with oncogenic signaling pathways. One of these key pathways, PI3K-AKT-mTOR, are crucial kinases that manipulate multiple cell functions, including proliferation and survival under cellular stress (Koundouros et al, 2020; Tewari et al, 2022; Vasan et al, 2022). An array of cancers harboring oncogenic alterations of PIK3CA, KRAS, PTEN, EGFR, and RHOA have been reported to be associated with activated PI3K/AKT-mTOR signaling (Gouw et al, 2017; Kumagai et al, 2020; Sugiyama et al, 2020a; Sun et al, 2021; Tewari et al, 2022). A previous investigation uncovered that RHOA Y42C mutation, a driver mutation that accounts for 10–25% of diffuse-type GC, could develop PD-1 blockade resistance by stimulating PI3K-AKT-mTOR-FAS signaling (Kumagai et al, 2020). Members of the ARHGAP family are regulatory factors of the RHOA pathway, which could inactivate RHOA by interacting with GAP domains and stimulating the GTPase activities of RHOA. Given that the CLDN18-ARHGAP fusion protein retains the GAP domain of ARHGAP, it may still be able to regulate the RHOA pathway (Yao et al, 2015). Our study revealed that GC with CLDN18-ARHGAP fusion could also activate PI3K-AKT-mTOR-FAS to enhance FFA production and promote Treg cells

proliferation and survival in the TME by means of bioinformatic analysis, in vitro and in vivo experiments. Pharmaceutically leveraging abnormally activated PI3K pathway could inhibit tumor growth and boost anti-tumor immune responses by facilitating immune infiltration or blocking the effects of immunosuppressive cells (Okkenhaug et al, 2016; Sun et al, 2021). We found that PI3Ki not only repressed tumor growth in GCs with CLDN18-ARHGAP fusion but also restored the TIME with increased CD8[+] T-cell infiltration. Collectively, our results suggested that metabolic advantages caused by driver gene mutations may be a common factor in forming an immunosuppressive TIME.

Due to technical limitations, there remain deficiencies in our current investigation of immunotherapy targeting CLDN18-ARHGAP gene fusion. Firstly, tumors with RHOA Y42C mutation, FAK, PTEN, EGFR, and WNT-β-catenin alterations could modulate the chemokine milieu in the TIME to inhibit CD8[+] T-cell infiltration (Jiang et al, 2016; Kumagai et al, 2020; Peng et al, 2016; Sugiyama et al, 2020b). Thus, the chemokine milieu in the TIME of GCs with CLDN18-ARHGAP gene

fusion may be worth further study. Also, we lacked data on the combined effects of NRT cells and PI3Ki without suitable animal models of humanized mice harboring HLA-A*11:01. Moreover, to avoid rejection reaction, PBMCs from the same donors should be used to build humanized mice and induce NRTs. As a result, it's impossible for us to finish this experiment as we need 1000 milliliters of blood from the same donor.

To summarize, our study is a pioneer in exploiting clinical applications of immunotherapies for GC with CLDN18-ARHGAP gene fusion. We are the first to identify immunogenic neoantigens derived from the CLDN18-ARHGAP gene fusion and verify the anti-tumor ability of NRT cells. Also, we reveal the role of CLDN18-ARHGAP gene fusion in promoting a suppressive TIME by facilitating the survival of Treg cells via a mechanism involving PI3K-AKT-mTOR-FAS activation and FFA production. These results may provide promising immunotherapies for gastric cancer patients with CLDN18-ARHGAP fusion gene through both immunogenic and immunoregulatory approaches (Fig. 7).

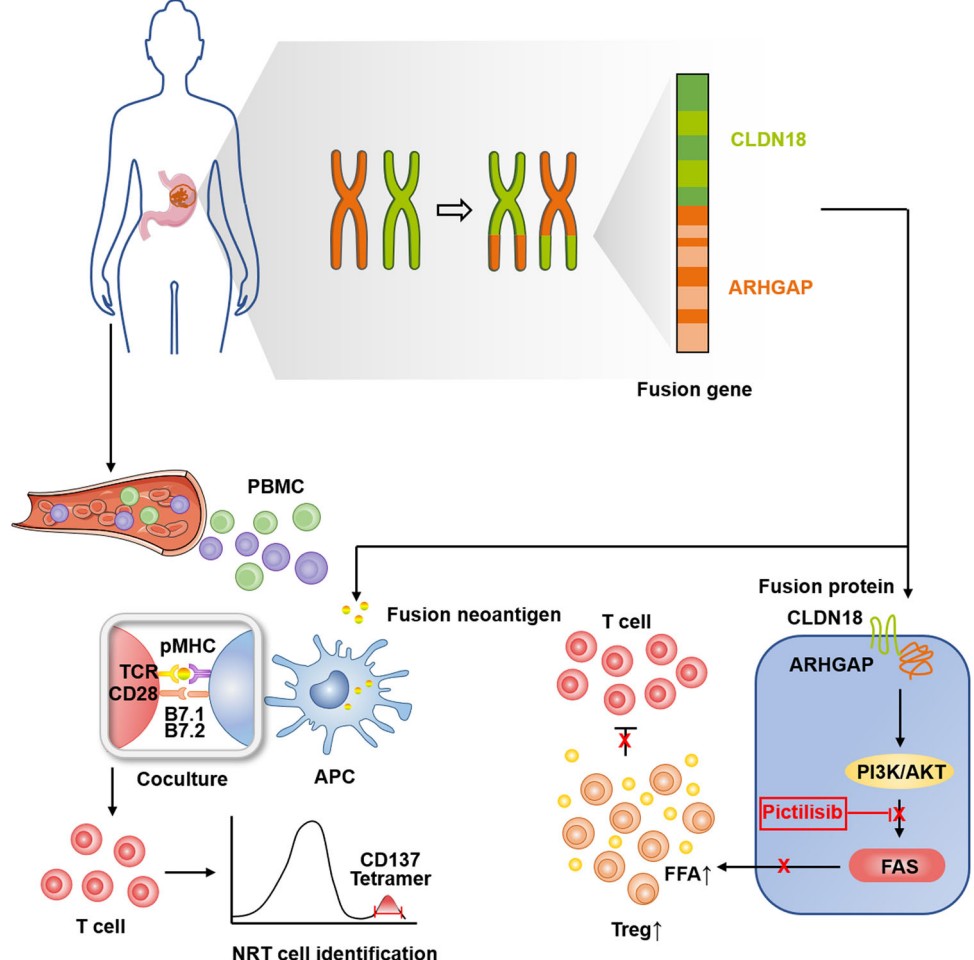

**Figure 7.  A proposed model for the immunotherapies targeting CLDN18-ARHGAP gene fusion in GC.**

CLDN18-ARHGAP gene fusion in GC generates an immunogenic neoantigen, which induces NRT cells specifically targeting and eliminating GC cells with CLDN18-ARHGAP fusion. Also, CLDN18-ARHGAP fusion can directly activate the downstream PI3K/AKT-mTOR-FAS signaling, which promotes FFA production from GC cells. The increased FFA accumulation in TIME leads to upregulated infiltration and prolonged survival of Treg cells, thus contributing to a suppressive TIME. Moreover, inhibition of the PI3K signaling can effectively reverse the upregulation of Treg cells and trigger tumor reduction. Therefore, NRT cells and/or PI3K inhibitors could be novel and promising immunotherapy for GC patients with CLDN18-ARHGAP fusion.

# Methods

### Reagents and tools table

| Reagent/resource | Reference or source | Identifier or catalog number |
|---|---|---|
| **Experimental models** | | |
| SNU601 (*H. sapiens*) | CO-BIOER | CBP60507, CVCL_0101 |
| MKN45 (*H. sapiens*) | CO-BIOER | CBP60488, CVCL_0434 |
| MFC (*M. musculus*) | CO-BIOER | CBP60882, CVCL_5J48 |
| 615-line mice (*M. musculus*) | Cavens Laboratory Animal Co., Ltd | M0026 |
| Balb/c nude mice (*M. musculus*) | Cavens Laboratory Animal Co., Ltd | C000103 |
| **Antibodies** | | |
| Anti-mouse CD45 (FITC) | Biolegend | #157607 |
| Anti-mouse CD3 (FITC) | Biolegend | #100203 |
| Anti-mouse CD3 (PC7) | Biolegend | #100320 |
| Anti-mouse CD4 (PE) | Biolegend | #100407 |
| Anti-mouse CD4 (FITC) | Biolegend | #100406 |
| Anti-mouse CD8 (APC) | Biolegend | #100711 |
| Anti-mouse CD8 (PC5.5) | Biolegend | #100734 |
| Anti-human/mouse CD11b (APC) | Biolegend | #101211 |
| Anti-mouse CD11c (FITC) | Biolegend | #117305 |
| Anti-mouse F4/80 (FITC) | Biolegend | #123107 |
| Anti-mouse CD83 (PE) | Biolegend | #121508 |
| Anti-mouse CD86 (PE) | Biolegend | #159203 |
| Anti-mouse CD163 (PE) | Biolegend | #156703 |
| Anti-mouse Ly-6G/Ly-6C (Gr-1) (PE) | Biolegend | #108407 |
| Anti-mouse I-A/I-E (MHC-II) (APC) | Biolegend | #107613 |
| Anti-AKT | Cell Signaling Technology | #9272 |
| Anti-p-AKT | Cell Signaling Technology | #4060 |
| Anti-mTOR | Cell Signaling Technology | #2983 |
| Anti-p-mTOR | Cell Signaling Technology | #5536 |
| Anti-Raptor | Cell Signaling Technology | #2280 |
| Anti-FAS | Cell Signaling Technology | #3180 |
| Anti-β-actin | Cell Signaling Technology | #3700 |
| Anti-SERBP-1 | Santa Cruz | #sc-13551 |
| **Oligonucleotides and other sequence-based reagents** | | |
| Plasmid | GENECHEM | N/A |
| **Chemicals, enzymes, and other reagents** | | |
| Peptides | ChinaPeptides | N/A |
| AIM-V medium | Gibco | 12055083 |
| Interleukin-2 | PeproTech | 200-02 |

| Reagent/resource | Reference or source | Identifier or catalog number |
|---|---|---|
| Interleukin-4 | PeproTech | 200-04 |
| Interleukin-7 | PeproTech | 200-07 |
| Interleukin-15 | PeproTech | 200-15 |
| GM-CSF | PeproTech | 300-23 |
| Lipopolysaccharide (LPS) | Sigma-Aldrich | L2630 |
| IFN-γ | PeproTech | 300-02 |
| Enzyme-linked immunospot assay (ELISPOT) | Dakewei | CT230-PR2 |
| IFN-γ Flex Set (Bead B8) | BD Biosciences | 560111 |
| Human TH1/TH2 | BioLegend | 741029 |
| CFSE | Invitrogen | C34554 |
| Pictilisib | Selleck | S1065 |
| BCA Protein Quantification Kit | Vazyme | E112-01 |
| Immobilon Western HRP Substrate | Merck | WBKLS0050 |
| Mouse Regulatory T Cell Staining Kit#2 | eBioscience | 88-8118 |
| Free Fatty Acid Quantification Kit | BioVision | K612-100 |
| **Software** | | |
| NetMHCpan 4.0 | https://services.healthtech.dtu.dk/services/NetMHC-4.0/ | |
| GraphPad Prism 8.0.1 | https://www.graphpad-prism.cn/ | |
| **Other** | | |
| NGS | OrigiMed | YuanSu panel assay |
| PCR-SBT | CapitalBio Technology | N/A |
| ELISPOT CTL Reader | Cellular Technology Inc. | N/A |

## Patients and samples

In total, 87 formalin-fixed paraffin-embedded (FFPE) tumor samples of GC patients from Drum Tower Hospital, Medical School of Nanjing University were collected by us between March 2019 and April 2021. Informed consent was obtained from all patients. All histopathologic diagnoses were reviewed by at least two senior pathologists independently. Clinical information was retrospectively collected. This study was conducted in accordance with the code of ethics of the World Medical Association (Declaration of Helsinki), the Department of Health and Human Services Belmont Report and approved by the Ethics Committee of Nanjing Drum Tower Hospital (No. 2021-324-01).

## Nucleic acid preparations and next-generation sequencing (NGS)

FFPE tissues obtained from patients with GC were collected for NGS in the YuanSu panel assay (OrigiMed, Shanghai). DNA was extracted from the unstained FFPE sections with tumor content of no less than 20% and was fragmented to ~250 bp by sonication. A library was constructed using the KAPA Hyper Prep Kit (KAPA Biosystems), and hybridization capture was performed with a custom panel containing individually synthesized 5'-biotinylated DNA probes. Paired-end sequencing was performed according to

the manufacturer's protocols. Genomic alterations, including SNV&INDELs, copy number variations (CNVs), and gene fusions, were assessed using the OrigiMed-pipeline. Genomic alterations relevant to cancer immunotherapy, which included TMB levels and MSI, were also evaluated. Mutation signature was predicted using a public software deconstuctSigs. All substitution mutations were classified into 96 kinds of trinucleotides, and the frequency of each was precisely calculated as the characteristic signature of these samples. The signature was then compared with the typical 30 signatures from COSMIC to identify the most similar combination and the percentage of each contribution. The datasets produced in this study are upload in the China National Center for Bioinformation (Chen et al, 2021; CNCB-NGDC Members and Partners, 2022).

## Cell lines

Human GC cell lines SNU601 (CO-BIOER CBP60507, CVCL_0101) and MKN45 (CO-BIOER CBP60488, CVCL_0434) and mouse GC cell line MFC (CO-BIOER CBP60882, CVCL_5J48) were cultured in Roswell Park Memorial Institute (RPMI) 1640, supplemented with 10% fetal calf serum (FBS), 100 U/mL penicillin, and 100 μg/mL streptomycin at 37 °C and 5% $CO_2$. The mycoplasma contamination was examined routinely using a PCR mycoplasma detection kit every month. Cell lines were recently authenticated.

## Overexpressing cell lines

The lentiviruses were purchased from GENECHEM. MFC, SNU601, and MKN45 cells were transfected with supernatants containing lentiviruses carrying constructs for the overexpression of CLDN18-ARHGAP fusion and empty vectors using the Lipofectamine 2000 reagent (Invitrogen). After 48 h, the infected cells were selected with 7 (for MFC) or 2 (for SNU601 and MKN45) mg/mL puromycin.

## Mice

615-line mice (catalog number: M0026, the syngeneic hosts of the MFC GC cell line) and the Balb/c nude mice (catalog number: C000103, T-cell deficient) were purchased from Cavens Laboratory Animal Co., Ltd (Changzhou, China) and housed in a specific pathogen-free facility with a relative humidity of 55% ± 5%, room temperature of 22 °C ± 2 °C, and 12-h/12-h light/dark cycle. All animals were maintained in the pathogen-free animal facilities at Nanjing University Medical School Affiliated Drum Tower Hospital (Nanjing, China). In all, 4–6-week-old female or male mice were randomized based on age and weight and used for all in vivo experiments. All animal experiments were approved by the Institutional Animal Care and Use Committee of Drum Tower Hospital (approval number: 2020AE01068).

## Human leukocyte antigen (HLA) typing

Four-digit HLA class I alleles (HLA-A) were identified by PCR-sequence-based typing (PCR-SBT) on patients or healthy donor peripheral blood (CapitalBio Technology, Beijing, China).

## Epitope prediction and peptide synthesis

The chimeric amino acid sequence of the CLDN18-ARHGAP fusion gene was queried using NetMHCpan 4.0 tools to predict major histocompatibility complex (MHC) class I binding 9- to 11-mer fusion peptides to the patients' HLA-A. The lower the %rank, the better the immunogenicity of the peptide. Fusion peptides with good immunogenicity were tested for their ability to stimulate patients' or healthy donors' T cells to secret interferon-γ (IFN-γ). Wild-type peptides in the corresponding position of CLDN18 and ARHGAP gene were tested as a control. Customized peptides were obtained from ChinaPeptides (Shanghai, China) and 0.5% dimethylsulfoxide (DMSO) may be added to help dissolve.

## Analysis of T-cell response

Patients' and healthy donors' autologous peripheral blood mononuclear cells (PBMCs) were used to evaluate the immunogenicity of candidate neoantigens in vitro. An established simple and effective culture protocol was used in detecting and monitoring NRT cells. Briefly, heparinized blood samples were obtained from GC patients with CLDN18-ARHGAP fusion and healthy donors for the isolation of PBMCs by centrifugation on a Ficoll density gradient and suspended in AIM-V medium (Gibco). In each U-bottomed well, $1 \times 10^5$ PBMCs were incubated with a corresponding peptide (25 μM) in 200 μl culture medium, which was applied to facilitate cell-to-cell contact. The culture medium consisted of AIM-V medium, 10% FBS (Gibco), and interleukin-2 (IL-2, 100 U/ml; PeproTech). For peptide stimulation at 3-day intervals, half of the culture medium containing a corresponding peptide (25 μM) and IL-2 (100 U/ml) was changed. After 3 cycles of peptide stimulation followed by an overnight restimulation, on day 10, the specific T-cell responses to each peptide were evaluated by enzyme-linked immunospot assay (ELISPOT) and Capture Bead Assay (CBA). Recognition of fusion neoantigens was tested as compared with wild-type peptide control and negative (DMSO) control. In addition, the T-cell activation marker CD137 (4-1BB) was assessed by flow cytometry.

## IFN-γ ELISPOT assay

IFN-γ ELISPOT kit (CT230-PR2, Dakewei) was used to determine the frequency of cytokine-secreting T cells after overnight activation with peptide. Briefly, peptide-stimulated PBMCs or dentric cell (DC)-pulsed peptide coculture with T cells ($10^5$ per well) were added to duplicate wells for 18–20 h. The plates were washed before the addition of the diluted detection antibody (1:100 dilution) and then incubated for 1 h at 37 °C. After washing the plates, streptavidin–horseradish peroxidase (HRP) (1:100 dilution) was added and incubated at 37 °C for another 1 h. 3-Amino-9-ethylcarbazole (AEC) solution mix was then added to each well, and the plates were left in the dark for about 15–25 min at room temperature before deionized water was added to stop development. Plates were scanned by ELISPOT CTL Reader (Cellular Technology Inc.), and the results were analyzed with ELISPOT software (AID). Spots greater than twice the no-peptide (media) control were considered positive for T-cell reactivity.

## Cytometric bead array analysis of cytokines

The concentrations of cytokines in culture supernatants were measured by cytometric bead array according to the manufacturer's protocol with an appropriate diluent. Human IFN-γ Flex Set (Bead B8) (560111, BD Biosciences) was used for detection of single-cytokine IFN-γ, and Human TH1/TH2 (741029, BioLegend) was used for the detection of tumor necrosis factor-α (TNF-α), IFN-γ, IL-2, interleukin-4 (IL-4), interleukin-5 (IL-5), interleukin-6 (IL-6), interleukin-10 (IL-10), and interleukin-13 (IL-13).

## Generation of DCs and neoantigen-reactive T cells

Monocyte-derived DCs were generated by plate adherence of PBMCs. Briefly, PBMCs were set to $5 \times 10^6$ to $10^7$ cells/ml in AIM-V medium and incubated for 2 h at 37 °C, 5% carbon dioxide ($CO_2$). Then, nonadherent cells were collected and washed. The adherent cells were cultured for 72 h with CellGro DC media (CellGenix) containing 1% human serum (HS; collected and processed in-house), granulocyte-macrophage colony-stimulating factor (GM-CSF, 800 IU/ml), and IL-4 (1000 IU/ml). The immature DCs were then lifted and resuspended in fresh medium containing 1% HS, GM-CSF (800 IU/ml), IL-4 (1000 IU/ml), lipopolysaccharide (LPS, 10 ng/ml), and IFN-γ (100 IU/ml) (LPS from Sigma-Aldrich, cytokines from PeproTech) and incubated for approximately 16–48 h. Mature DCs were pulsed with identified peptides (10 μM) individually for approximately 4–6 h at 37 °C, and washed with prewarmed phosphate buffer saline (PBS). Peptide-pulsed DCs were incubated with T cells at a ratio of 1:5 to 1:10 in complete AIM-V medium supplemented with 5% HS, IL-2 (100 U/ml), IL-7 (10 ng/ml), and IL-15 (10 ng/ml). The fresh complete medium containing cytokines was added every 2–3 days. On days 7–10, the proportion of NRT cells was assessed by flow cytometry or ELISPOT assays. According to the expansion of the NRT cells, the irradiated K562-based artificial antigen-presenting cells (APCs) or irradiated autologous PBMCs loading antigens were cocultured with T cells for restimulation. (the K562 cells expressing CD137L, CD80, and HLA-A*1101 were constructed by our laboratory).

## Cytotoxicity assay

The NRT cells or T cells without induction were tested for lytic activities by carboxyfluorescein succinimidyl ester/propidium iodide (CFSE/PI) labeling cytotoxicity assay. SNU601 and MKN45 cells expressing CLDN18-ARHGAP fusion were used as target cells. Target cells were labeled with 4 mM CFSE (Invitrogen) for 10 min at 37 °C in PBS. Labeling was stopped by adding of a tenfold volume of PBS and extensively washed in PBS. CFSE-labeled cells were then incubated with T cells at effector/target ratios of 10:1 for 6 h. PI (Sigma-Aldrich) was added to determine the ratio of cell death. Samples were analyzed by flow cytometry.

## Syngeneic mouse tumor models

For subcutaneous tumor models, MKN45 human GC cells ($1 \times 10^7$ per animal), and MFC mouse GC cells ($1 \times 10^6$ per animal) were suspended in PBS and subcutaneously (s.c.) injected into 4–6-week-old sex-matched Balb/c nude mice or 615-line mice to form solid tumors. For abdominal tumor models, MFC mouse GC cells

($3 \times 10^6$ per animal) were suspended in PBS and intraperitoneally (i.p.) injected into 4–6-week-old sex-matched 615-line mice to form solid tumors. In some studies, NRT or T cells were injected intravenously (i.v.) injected twice in 20 days. In some studies, Pictilisib (PI3K inhibitor, Selleck, S1065) or DMSO control were injected i.p. for seven times in 15 days. During an experiment, mouse body weight was recorded and tumor growth was monitored every two or three days by measuring tumor size using a digital caliper; tumor volumes were calculated by the formula $1/2 \times L \times W^2$. At the treatment endpoint of an experiment, mice were sacrificed and tumors, draining lymph nodes (LN), and spleens were removed for further analysis using IHC and/or Flow cytometry.

## Flow cytometry analysis

For immune cell analysis from different GC models in vivo, single-cell suspensions were prepared from tumor tissue, draining lymph nodes and spleen immediately upon mouse sacrifice. Draining lymph nodes and spleens were mechanically dissociated and single-cell suspensions were obtained through 40-μm nylon cell strainers (Biosharp). Tumor tissues were shredded, and then digested with 1 mg/ml collagenase IV (Sigma-Aldrich) and 100 U/ml DNase I (Sigma-Aldrich) in serum-free RMPI-1640 for 1 h at 37 °C incubator. Cells were filtered through 40-μm nylon cell strainers (Biosharp), and washed twice with PBS. Red blood cells were then removed from single-cell suspensions with red blood cell lysis buffer (Biosharp) and washed twice with PBS for the following antibody staining. To detect immune cell surface marker expression, single-cell suspensions were stained with anti-mouse CD45 (FITC, #157607), anti-mouse CD3 (FITC, #100203; PC7 #100320), anti-mouse CD4 (PE, #100407; FITC #100406), anti-mouse CD8 (APC, #100711; PC5.5 #100734), anti-human/mouse CD11b (APC, #101211), anti-mouse CD11c (FITC, #117305), anti-mouse F4/80 (FITC, #123107), anti-mouse CD83 (PE, #121508), anti-mouse CD86 (PE, #159203), anti-mouse CD163 (PE, #156703), anti-mouse Ly-6G/Ly-6C (Gr-1) (PE, #108407), anti-mouse I-A/I-E (MHC-II) (APC, #107613) from Biolegend, and Mouse Regulatory T Cell Staining Kit#2 (88-8118) from eBioscience. All these antibodies were used with 1:1000 dilution.

## Western blot analysis

For protein detection, tumor cells were cultured in vitro in six-well plates at $1 \times 10^6$ cells per well with or without Pictilisib (5 mM) for 48 h. The total protein concentrations of each sample were measured and quantified using BCA Protein Quantification Kit (Vazyme, E112-01). After that, 20 μg of total protein from each sample was resolved on a 10% SDS–polyacrylamide electrophoresis gel and electroblotted onto polyvinylidene difluoride (PVDF) membranes. Proteins were detected by incubation with 1:1000 or 1:2000 dilution of primary antibodies, washed and incubated with goat anti-rabbit HRP antibody (1:5000, Beyotime), and detected after incubation with Immobilon Western HRP Substrate (Merck, WBKLS0050). Primary antibodies directed against AKT (#9272, 1:1000), p-AKT (#4060, 1:2000), mTOR (#2983,1:1000), p-mTOR (#5536, 1:1000), Raptor (#2280, 1:1000), FAS (#3180, 1:1000) and β-actin (#3700, 1:1000) were from Cell Signaling Technology and SERBP-1 (#sc-13551, 1:200) was from Santa Cruz.

**The paper explained**

**Problem**

The CLDN18-ARHGAP fusion gene is an oncogenic driver newly discovered in gastric cancer which occurs frequently. However, there are still no effective therapies targeting it.

**Results**

The CLDN18-ARHGAP fusion gene-derived neoantigen has good immunogenicity, and neoantigen-reactive T cells induced by it have specific and robust anti-tumor capacity both in in vitro coculture models and in vivo xenograft gastric cancer models. Also, gastric cancer cell with CLDN18-ARHGAP fusion gene could activate PI3K/AKT-mTOR-FAS signaling, which enhances free fatty acid production of gastric cancer cells to favor the survival of Treg cells and contribute to immune suppression.

**Impact**

These results provide a promising novel approach for treating gastric cancer patients with CLDN18-ARHGAP fusion gene through both immunogenic and immunoregulatory approaches.

## Assay measuring the FFA concentration

To quantify the FFA concentration in the culture medium, all the cell lines were maintained in RPMI medium supplemented with 10% lipids without FBS. Interstitial fluids from tumors were collected by centrifugation as previously described [29]. The concentrations of total FFAs were assessed with the Free Fatty Acid Quantification Kit (BioVision, Zurich, Switzerland).

## Statistical analysis

GraphPad Prism 8.0.1 software were used for statistical analysis of data and graphic representations. We employed a random number table to perform randomization. All animal experiments in this study were performed and analyzed in a blinded manner. Treatment groups were assigned in a random fashion. Every mouse was assigned a temporary random number within the weight range. The mice were given their permanent numerical designation in the cages after they were randomly categorized into each group. For each group, a cage was selected randomly from the pool of all cages. All data were collected and analyzed by two blinded observers. No statistical methods were used to predetermine sample sizes. No analyzed samples were omitted from the report. For comparisons between two groups, unpaired $t$ test, log-rank test or Wilcoxon rank-sum test were used as indicated. The error bars of data were presented as the means ± standard error of the mean (SEM). The $P$ value of less than 0.05 was considered to be statistically significant. ns, not significant; $*P < 0.05$, $**P < 0.01$, $***P < 0.001$ and $****P < 0.0001$.

## Data availability

The datasets produced in this study are available in the following databases:- NGS-seq data: China National Center for Bioinformation HRA007573.

The source data of this paper are collected in the following database record: biostudies:S-SCDT-10_1038-S44321-024-00120-3.

## Peer review information

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

## Acknowledgements

The authors would like to thank OrigiMed, for the kind help of NGS. This work is supported by grants from National Natural Science Foundation of China (82073382), the Fundamental Research Funds for the Central Universities (0214-14380506), the Nanjing Health Technology Development General Project (YKK23084), Jiangsu Provincial Natural Science Foundation Youth Project (BK20230151) and Beijing Xisike Clinical Oncology Research Foundation (Y-Young2023-019).

## Author contributions

**Yue Wang**: Conceptualization; Formal analysis; Funding acquisition; Investigation; Writing—original draft. **Hanbing Wang**: Visualization; Writing—review and editing. **Tao Shi**: Formal analysis; Methodology. **Xueru Song**: Validation; Visualization. **Xin Zhang**: Investigation. **Yue Zhang**: Investigation. **Xuan Wang**: Investigation. **Keying Che**: Methodology. **Yuting Luo**: Investigation. **Lixia Yu**: Supervision. **Baorui Liu**: Supervision. **Jia Wei**: Conceptualization; Supervision; Writing—review and editing.

Source data underlying figure panels in this paper may have individual authorship assigned. Where available, figure panel/source data authorship is listed in the following database record: biostudies:S-SCDT-10_1038-S44321-024-00120-3.

## Disclosure and competing interests statement

The authors declare no competing interests.

