## [Peer Review File · EMBO Molecular Medicine]

Immunotherapies targeting the oncogenic fusion gene CLDN18-ARHGAP in gastric cancer

Yue Wang, Hanbing Wang, Tao Shi, Xueru Song, Xin Zhang, Yue Zhang, Xuan Wang, Keying Che, Yuting Luo, Lixia Yu, Baorui Liu, and Jia Wei

Corresponding author: Jia Wei (jiawei99@nju.edu.cn)

Review Timeline:

Submission Date:	28th Aug 23
Editorial Decision:	17th Oct 23
Revision Received:	23rd Feb 24
Editorial Decision:	13th Mar 24
Revision Received:	10th Jun 24
Editorial Decision:	1st Jul 24
Revision Received:	31st Jul 24
Accepted:	2nd Aug 24

Editor: Poonam Bheda

Transaction Report:

17th Oct 2023

Dear Dr. Wei,

Thank you for the submission of your manuscript to EMBO Molecular Medicine. We have now received feedback from the three reviewers who agreed to evaluate your manuscript. As you will see from the reports below, the referees acknowledge the interest of the study and are overall supporting publication of your work pending appropriate revisions.

Addressing the reviewers' concerns in full will be necessary for further considering the manuscript in our journal, and acceptance of the manuscript will entail a second round of review. EMBO Molecular Medicine encourages a single round of revision only and therefore, acceptance or rejection of the manuscript will depend on the completeness of your responses included in the next, final version of the manuscript. For this reason, and to save you from any frustrations in the end, I would strongly advise against returning an incomplete revision.

We are expecting your revised manuscript within three months, if you anticipate any delay, please contact us.

We require:

4) A .docx formatted letter INCLUDING the reviewers' reports and your detailed point-by-point responses to their comments. As part of the EMBO Press transparent editorial process, the point-by-point response is part of the Review Process File (RPF), which will be published alongside your paper.

5) A complete author checklist, which you can download from our author guidelines (<https://www.embopress.org/page/journal/17574684/authorguide#submissionofrevisions>). Please insert information in the checklist that is also reflected in the manuscript. The completed author checklist will also be part of the RPF.

6) Please note that all corresponding authors are required to supply an ORCID ID for their name upon submission of a revised manuscript.

7) It is mandatory to include a 'Data Availability' section after the Materials and Methods. Before submitting your revision, primary datasets produced in this study need to be deposited in an appropriate public database, and the accession numbers and database listed under 'Data Availability'. Please remember to provide a reviewer password if the datasets are not yet public (see <https://www.embopress.org/page/journal/17574684/authorguide#dataavailability>).

In case you have no data that requires deposition in a public database, please state so in this section. Note that the Data Availability Section is restricted to new primary data that are part of this study. This study includes no data deposited in external repositories.

8) For data quantification: please specify the name of the statistical test used to generate error bars and P values, the number (n) of independent experiments (specify technical or biological replicates) underlying each data point and the test used to calculate p-values in each figure legend. The figure legends should contain a basic description of n, P and the test applied. Graphs must include a description of the bars and the error bars (s.d., s.e.m.). Please provide exact p values.

9) Our journal encourages inclusion of *data citations in the reference list* to directly cite datasets that were re-used and obtained from public databases. Data citations in the article text are distinct from normal bibliographical citations and should directly link to the database records from which the data can be accessed. In the main text, data citations are formatted as

follows: "Data ref: Smith et al, 2001" or "Data ref: NCBI Sequence Read Archive PRJNA342805, 2017". In the Reference list, data citations must be labeled with "[DATASET]". A data reference must provide the database name, accession number/identifiers and a resolvable link to the landing page from which the data can be accessed at the end of the reference. Further instructions are available at .

13) Author contributions: CRediT has replaced the traditional author contributions section because it offers a systematic machine readable author contributions format that allows for more effective research assessment. Please remove the Authors Contributions from the manuscript and use the free text boxes beneath each contributing author's name in our system to add specific details on the author's contribution. More information is available in our guide to authors.

Please also suggest a striking image or visual abstract to illustrate your article as a PNG file 550 px wide x 300-600 px high. Share synopsis text and image, as well as eTOC:

Please note that these would be the final versions and changes during proofing are usually not allowed

16) As part of the EMBO Publications transparent editorial process initiative (see our Editorial at <http://embomolmed.embopress.org/content/2/9/329>), EMBO Molecular Medicine will publish online a Review Process File (RPF) to accompany accepted manuscripts.

In the event of acceptance, this file will be published in conjunction with your paper and will include the anonymous referee reports, your point-by-point response and all pertinent correspondence relating to the manuscript. Let us know whether you agree with the publication of the RPF and as here, if you want to remove or not any figures from it prior to publication. Please note that the Authors checklist will be published at the end of the RPF.

I look forward to receiving your revised manuscript.

Yours sincerely,

Poonam Bheda

Poonam Bheda, PhD
Scientific Editor
EMBO Molecular Medicine

***** Reviewer's comments *****

Referee #1 (Comments on Novelty/Model System for Author):

Most of the convincing data are only obtained in one gastric cancer xenograft (MKN45-OE tumors GC xenograft - overexpressing the CLDN18-ARHGAP fusion gene) and one syngeneic system (mouse forestomach carcinoma (MFC)) system e.g. Results of Western Blot analysis in Figure 5F is supposed to show that compared to MKN45/SNU601-NC (control plasmid) cells, the expressions of p-AKT, etc. were remarkably upregulated in MKN45/SNU601-OE cells (Fig. 5F), a closer examination of the blots (which have no cumulative statistical analysis from N = ? repeats) shows p-Akt was only really upregulated in the MKN45 not SNU601. How representative is this phenomenon (and its PI3K dependency) among a heterogeneous patient population (one of two cell lines currently in this manuscript - ideally more cell lines ?N=5 should be used to get a true estimate of the proportion of patients who may respond in this way. A second example is that MFC-OE group had significantly increased infiltrations of Tregs within both tumors and draining lymphocyte nodes (Fig. 4A, C). The statistical significance is marginal/borderline so again another syngeneic model is needed to reproduce this marginal statistical difference.

Referee #1 (Remarks for Author):

This is a conceptually interesting paper. The authors reported that CLDN18-ARHGAP fusions can generate immunogenic neoantigen peptides to induce neoantigen-reactive T (NRT) cells with robust and specific anti-tumor cytotoxicity. The authors went beyond just reporting on the phenomenon and demonstrated mechanistically that gastric cancer (GC) cells with CLDN18-ARHGAP fusion have increased free fatty acid (FFA) production caused potentially by activation of PI3K-AKT-mTOR/FAS signaling, thus promoting the infiltration and survival of Treg cells within the tumor immune microenvironment.

Although the mechanistic concepts are interesting and clearly the therapeutic tool described could be of utility clinically in this devastating disease, some significant issues exist and need to be resolved before publication:

1. Most of the convincing data are only obtained in one gastric cancer xenograft (MKN45-OE tumors GC xenograft - overexpressing the CLDN18-ARHGAP fusion gene) and one syngeneic system (mouse forestomach carcinoma (MFC)) system e.g. Results of Western Blot analysis in Figure 5F is supposed to show that compared to MKN45/SNU601-NC (control plasmid) cells, the expressions of p-AKT, etc. were remarkably upregulated in MKN45/SNU601-OE cells (Fig. 5F), a closer examination of the blots (which have no cumulative statistical analysis from N = ? repeats) shows p-Akt was only really upregulated in the MKN45 not SNU601 (OE vs NC). How representative is this phenomenon (and its PI3K dependency) among a heterogeneous patient population (one of two cell lines currently in this manuscript - ideally more cell lines ?N=5 should be used to get a true estimate of the proportion of patients who may behave in this way. A second example is that MFC-OE group had significantly increased infiltrations of Tregs within both tumors and draining lymphocyte nodes (Fig. 4A, C). The statistical significance is marginal/borderline so again another syngeneic model is needed to reproduce this marginal statistical difference (p=0.038 primary; p=0.0347 draining LN).

2. Out of four peptides, RTEDEVYNSNK (peptide 1, P1) derived from CLDN18 (exon5)-ARHGAP26 (exon12) fusion induced dramatically increased secretion of interferon- γ (IFN- γ) of more than 2 two folds (Fig. 2A, B). How many GC patients' autologous PBMCs with HLA-A*11:01 were presented here? Once again, how representative is this phenomenon (responsiveness to PI) among a heterogeneous patient population would be expected? There were 3 data points in 2B for P1; one was significantly increased than the other two in terms of IFN response - so are we expecting 1 in 3 patient responsiveness? Generally the legend is really not detailed so this conclusion is really according to interpretation by the reviewer. The authors should significantly increase the content of the legend to help readers to understand the results.

3. Going back to the finding that MFC-OE group had significantly (albeit marginal) increased (p=0.038 primary; p=0.0347 draining LN) infiltrations of Tregs within both tumors and draining lymphocyte nodes. Besides increasing the number of syngeneic tumor model beyond N=1, does this marginal statistical significance translate functionally to less granzyme positive effector T cells (IL2R+ among FOXP3 negative cells) in the tissues within the tumor and lymph nodes?

4. The authors reported an interesting phenomenon that free fatty acid (FFA) concentrations in the culture supernatants of MKN45-OE, SNU601-OE and MFC-OE cells (Fig. 5A, B) were increased in comparison to the control plasmid cells. What is the degree of unsaturation in these fatty acids? If polyunsaturated, then the authors should investigate whether there is sign of

ferroptosis (see Zeng K, Li W, Wang Y, Zhang Z, Zhang L, Zhang W, et al. Inhibition of CDK1 Overcomes Oxaliplatin Resistance by Regulating ACSL4-mediated Ferroptosis in Colorectal Cancer. *Adv Sci (Weinh)*. 2023:e2301088.) A few cell/tissue markers can be used for assessing ferroptosis see Chen X, Comish PB, Tang D, Kang R. Characteristics and Biomarkers of Ferroptosis. *Front Cell Dev Biol*. 2021 Jan 21;9:637162. doi: 10.3389/fcell.2021.637162. PMID: 33553189; PMCID: PMC7859349. Also, Feng H, Schorpp K, Jin J, Yozwiak CE, Hoffstrom BG, Decker AM, et al. Transferrin Receptor Is a Specific Ferroptosis Marker. *Cell Rep*. 2020;30(10):3411-23 e7 as well as Jin J, Schorpp K, Samaga D, Unger K, Hadian K, Stockwell BR. Machine Learning Classifies Ferroptosis and Apoptosis Cell Death Modalities with TfR1 Immunostaining. *ACS Chem Biol*. 2022;17(3):654-60.

Ferroptosis is increasingly recognized to be important in the field of tumor immunotherapy and this additional aspect should be explored if polyunsat. FA containing lipids are involved in this novel mechanism that the authors have found.

Referee #2 (Comments on Novelty/Model System for Author):

It is reasonable to detect anti-tumor capacity both in in vitro co-culture models and in vivo xenograft gastric cancer models.

Referee #2 (Remarks for Author):

This manuscript focuses on gene fusion neoantigen as an effective cancer target, the authors first identified a recurrent fusion gene CLDN18-ARHGAP in Chinese gastric cancer cohorts, then they examined the immunogenic potential through in vitro co-culture models and in vivo xenograft gastric cancer models. Finally, the key mechanism how CLDN18-ARHGAP regulated tumor pathways and tumor immune microenvironment were explored. The design of the study is reasonable and the manuscript is written well, I only have one important concern about how the authors select CLDN18-ARHGAP as their investigation target. The mutation landscape of gastric cancer cohort only include CLDN18-ARHGAP related fusion gene, how about other fusion gene? The authors should clarify it.

Referee #3 (Comments on Novelty/Model System for Author):

This is an interesting study, which is focused on an potential important target of gastric cancers. Although the CLDN18-ARHGAP fusion has been identified and investigate for some years. There is no substantial progress in the treatment of this fusion gene. This study given us a new sight for the treatment of GC patients with CLDN18-ARHGAP fusion.

Referee #3 (Remarks for Author):

1. The landscape of mutations should be compared with other studies. Notably, the heatmap does not illustrate any copy number deletion, and the reasons for selecting sCNA genes should be addressed.
2. In some experiment designs, three mice per group are not sufficient to fully evaluate the results. At least five mice in each group should be used.
3. Although the regulatory phenomenon of CLDN18-ARHGAP on PI3K/Akt signaling has been validated, the underlying regulatory mechanism has not been discussed or investigated.
4. Abbreviations should be defined at their first mention in the manuscript.
5. Fig1: $\log(\text{TMB}+1)$ why plus 1, is there sample carried no mutation? what's the base of log? No Rearrangement was observed. what's the definition of shortindel and longindel?
6. Fig3b: no error bar in the line chart.
7. There are several typos present in the manuscript that should be corrected. eg: In "loss or amplification of driver genes like PIK3CA, PETN, or AKT": PETN should be PTEN; In "in 2018, Yang et al. identified a frequent proportion of CLDN18-ARHGAP fusion gene up to 25% in gastric signet ring cell carcinoma (SRCC)": Yang et al. should be Shu et al.

Jan 29, 2024

Prof. Poonam Bheda
Scientific Editor
EMBO Molecular Medicine

Re: EMM-2023-18597

Immunotherapies targeting the oncogenic fusion gene CLDN18-ARHGAP in gastric cancer.

Dear Prof. Poonam Bheda:

We sincerely appreciate your efforts in organizing the review process of our manuscript (EMM-2023-18597) and the opportunity for our manuscript to be under consideration with revisions. We also would like to thank the three reviewers for their professional and constructive comments on our manuscript. In response to these comments and suggestions, we have made very detailed and extensive revisions to the original manuscript, and all the concerns raised by the three reviewers have been answered and addressed by us in the point-to-point response letter and the revised manuscript.

Referee #1 (Comments on Novelty/Model System for Author):

Most of the convincing data are only obtained in one gastric cancer xenograft (MKN45-OE tumors GC xenograft - overexpressing the CLDN18-ARHGAP fusion gene) and one syngeneic system (mouse forestomach carcinoma (MFC)) system e.g. Results of Western Blot analysis in Figure 5F is supposed to show that compared to MKN45/SNU601-NC (control plasmid) cells, the expressions of p-AKT, etc. were remarkably upregulated in MKN45/SNU601-OE cells (Fig. 5F), a closer examination of the blots (which have no cumulative statistical analysis from N= ? repeats) shows p-Akt was only really upregulated in the MKN45 not SNU601 (OE vs NC). How representative is this phenomenon (and its PI3K dependency) among a heterogeneous patient population (one of two cell lines currently in this manuscript – ideally more cell lines? N=5 should be used to get a true estimate of the proportion of patients who may behave in this way. A second example is that MFC-OE group had significantly increased infiltrations of Tregs within both tumors and draining lymphocyte nodes (Fig. 4A, C).

The statistical significance is marginal/borderline so again another syngeneic model is needed to reproduce this marginal statistical difference ($p=0.038$ primary; $p=0.0347$ draining LN).

Response: Thanks a lot for your constructive and professional comments regarding the novelty and model system of our study. We totally agree with your comments and provide a point-to-point response based on your suggestions in the following comment 1. We also hope that our revisions can address your concerns.

Referee #1 (Remarks for Author):

This is a conceptually interesting paper. The authors reported that CLDN18-ARHGAP fusions can generate immunogenic neoantigen peptides to induce neoantigen-reactive T (NRT) cells with robust and specific anti-tumor cytotoxicity. The authors went beyond just reporting on the phenomenon and demonstrated mechanistically that gastric cancer (GC) cells with CLDN18-ARHGAP fusion have increased free fatty acid (FFA) production caused potentially by activation of PI3K-AKT-mTOR/FAS signaling, thus promoting the infiltration and survival of Treg cells within the tumor immune microenvironment.

Although the mechanistic concepts are interesting and clearly the therapeutic tool described could be of utility clinically in this devastating disease, some significant issues exist and need to be resolved before publication:

Comment 1. Most of the convincing data are only obtained in one gastric cancer xenograft (MKN45-OE tumors GC xenograft - overexpressing the CLDN18-ARHGAP fusion gene) and one syngeneic system (mouse forestomach carcinoma (MFC)) system e.g. Results of Western Blot analysis in Figure 5F is supposed to show that compared to MKN45/SNU601-NC (control plasmid) cells, the expressions of p-AKT, etc. were remarkably upregulated in MKN45/SNU601-OE cells (Fig. 5F), a closer examination of the blots (which have no cumulative statistical analysis from $N=?$ repeats) shows p-Akt was only really upregulated in the MKN45 not SNU601 (OE vs NC). How representative is this phenomenon (and its PI3K dependency) among a heterogeneous patient population (one of two cell lines currently in this manuscript – ideally more cell lines? $N=5$ should be used to get a true estimate of the proportion of patients who may behave in this way. A second example is that MFC-OE group had significantly increased infiltrations of Tregs within both tumors and draining lymphocyte nodes (Fig. 4A, C). The statistical significance is marginal/borderline so again another syngeneic model is

needed to reproduce this marginal statistical difference ($p=0.038$ primary; $p=0.0347$ draining LN).

Response: Thank you for your constructive questions. We fully agree with you that repeating the experiment is necessary for a solid estimate. We have repeated Western Blot at least 5 times and show all the original blots below. It demonstrated that pAKT is also remarkably upregulated in SNU601-OE cells compared to SNU601-NC cells, and PI3Ki could reverse the activation. We also replaced the representative blot of pAKT expression of SNU601-NC/OE cells in our manuscript (Fig. 5F).

As MFC is the only commercial murine gastric cancer cell line that could be bought from biotechnology companies, we added an abdominal metastasis model using the MFC-OE/NC cell line instead of a new cell line (Fig. 6G-J, Appendix Fig. S6B). It showed that Tregs in the tumor immune microenvironment and tumor draining lymphnodes were significantly upregulated in MFC-OE compared to MFC-NC ($p < 0.0001$ primary, $p=0.0160$ draining LN, Fig. 6J, I). Additionally, only in MFC OE tumors could PI3Ki inhibit the tumor growth ($p=0.0275$, Fig. 6H), and downregulate Tregs both in tumor immune microenvironment and tumor draining lymphnodes ($p=0.0151$ primary, $p=0.0469$ draining LN, Fig. 6J, I).

Thanks again for your kind and helpful advice, and we sincerely hope our explanations and revisions can address your concerns.

Comment 2. Out of four peptides, RTEDEVYNSNK (peptide 1, P1) derived from CLDN18 (exon5)-ARHGAP26 (exon12) fusion induced dramatically increased secretion of interferon- γ (IFN- γ) of more than 2 two folds (Fig. 2A, B). How many GC patients' autologous PBMCs with HLA-A*11:01 were presented here? Once again, how representative is this phenomenon (responsiveness to P1) among a heterogeneous

patient population would be expected? There were 3 data points in 2B for P1; one was significantly increased than the other two in terms of IFN response - so are we expecting 1 in 3 patient responsiveness? Generally the legend is really not detailed so this conclusion is really according to interpretation by the reviewer. The authors should significantly increase the content of the legend to help readers to understand the results.

Response: Thanks for your careful review. We presented the results of one patient with CLDN18 (exon5)-ARHGAP26 (exon12) fusion in Fig. 2A, B, and C, while another healthy donor's results in Fig. 2D, E, F, and G. The 3 data points in 2B represent 3 independent samples from this patient. We stimulated the PBMCs separately, so it would present different results, as one was significantly increased than the other two in terms of IFN response. This is a common phenomenon when detecting the immunogenicity of peptides [1, 2]. And, in Fig. 2D and 2E, the healthy donor's T cells were used to detect the IFN response. It showed that after 2-round stimulation, the 3 independent samples demonstrated similar IFN response to P1, reaching up to 60-fold changes. In all, we concluded that peptide derived from CLDN18 (exon5)-ARHGAP26 (exon12) fusion gene could stimulate T cell response both in patients and healthy donors, showing potent immunogenicity.

Thanks again for your kind suggestions. We have modified the content of the legend of Fig. 2 and highlighted it in yellow.

References:

- [1] Deniger DC, Pasetto A, Robbins PF, et al., T-cell Responses to TP53 "Hotspot" Mutations and Unique Neoantigens Expressed by Human Ovarian Cancers. *Clin Cancer Res.* 2018 Nov 15;24(22):5562-5573.
- [2] Parkhurst MR, Robbins PF, Tran E, et al., Unique Neoantigens Arise from Somatic Mutations in Patients with Gastrointestinal Cancers. *Cancer Discov.* 2019 Aug;9(8):1022-1035.

Comment 3. Going back to the finding that MFC-OE group had significantly (albeit marginal) increased ($p=0.038$ primary; $p=0.0347$ draining LN) infiltrations of Tregs within both tumors and draining lymphocyte nodes. Besides increasing the number of syngeneic tumor model beyond $N=1$, does this marginal statistical significance translate functionally to less granzyme positive effector T cells (IL2R+ among FOXP3 negative cells) in the tissues within the tumor and lymph nodes?

Response: Thank you for your constructive questions and suggestions. We fully agree that repeating the experiments in another syngeneic tumor model is necessary to

validate our finding that MFC-OE leads to increased intratumoral Treg infiltration. However, MFC is the only commercial murine gastric cancer cell line that could be bought from biotechnology companies. Therefore, we established the MFC-challenged syngeneic peritoneum metastasis GC model instead of a new cell line. We found that MFC-OE also results in significant upregulation of Tregs with the tumor immune microenvironment and tumor draining lymphnodes ($p < 0.0001$ primary, $p = 0.0160$ draining LN, added as Fig. 6J, I). What's more, we totally acknowledge that Tregs have essential impacts on the function of effector T cells, by hampering their expression of important functional markers, such as GZMB, IL2R, and IFN- γ [1-3]. In our study, we observed in vitro that the p1 peptide derived from CLDN18-ARHGAP26 fusion induced obviously increased IFN- γ , TNF- α , and IL-6 secretion from T cells (Fig. 2). Meanwhile, we observed in vivo that PI3K inhibition-induced downregulation of Treg infiltration was accompanied with increased CD8⁺ T cell infiltration in the mouse GC model (Fig. 6F). Therefore, we also speculate that Treg upregulation could hamper T cell-mediated anti-tumor immunity based on aforementioned evidence, and PI3K inhibition is a promising immunotherapy strategy for GC patients with CLDN18-ARHGAP26 fusion. Thanks again for your comments, and we hope our revisions can address your concern.

References:

- [1] Munn D H, Sharma M D, Johnson T S. Treg Destabilization and Reprogramming: Implications for Cancer Immunotherapy [J]. *Cancer research*, 2018, 78(18): 5191-9.
- [2] Tanaka A, Sakaguchi S. Regulatory T cells in cancer immunotherapy [J]. *Cell Res*, 2017, 27(1): 109-18.
- [3] Wing J B, Tanaka A, Sakaguchi S. Human FOXP3(+) Regulatory T Cell Heterogeneity and Function in Autoimmunity and Cancer [J]. *Immunity*, 2019, 50(2): 302-16.

Comment 4. The authors reported an interesting phenomenon that free fatty acid acid (FFA) concentrations in the culture supernatants of MKN45-OE, SNU601-OE and MFC-OE cells (Fig. 5A, B) were increased in comparison to the control plasmid cells. What is the degree of unsaturation in these fatty acids? If polyunsaturated, then the authors should investigate whether there is sign of ferroptosis (see Zeng K, Li W, Wang Y, Zhang Z, Zhang L, Zhang W, et al. Inhibition of CDK1 Overcomes Oxaliplatin Resistance by Regulating ACSL4-mediated Ferroptosis in Colorectal Cancer. *Adv Sci*

(Weinh). 2023:e2301088.) A few cell/tissue markers can be used for assessing ferroptosis see Chen X, Comish PB, Tang D, Kang R. Characteristics and Biomarkers of Ferroptosis. *Front Cell Dev Biol.* 2021 Jan 21; 9:637162. doi:10.3389/fcell.2021.637162. PMID: 33553189; PMCID: PMC7859349. Also, Feng H, Schorpp K, Jin J, Yozwiak CE, Hoffstrom BG, Decker AM, et al. Transferrin Receptor Is a Specific Ferroptosis Marker. *Cell Rep.* 2020;30(10):3411-23 e7 as well as Jin J, Schorpp K, Samaga D, Unger K, Hadian K, Stockwell BR. Machine Learning Classifies Ferroptosis and Apoptosis Cell Death Modalities with TfR1 Immunostaining. *ACS Chem Biol.* 2022;17(3):654-60.

Ferroptosis is increasingly recognized to be important in the field of tumor immunotherapy and this additional aspect should be explored if polyunsat. FA containing lipids are involved in this novel mechanism that the authors have found.

Response: We are very grateful for your kind suggestions and detailed references, and fully agree that ferroptosis may be an important mechanism underlying the tumor immune suppression caused by CLDN18-ARHGAP26 fusion. Based on your suggestions, we detected supernatants of tumor cells *in vitro* and found that the overexpression (OE) of CLDN18-ARHGAP26 fusion in GC cells did not affect the unsaturation degree of fatty acids (shown in the figure below). Moreover, we observed that there were no significant changes in the relative intracellular Fe²⁺ expression among GC cell lines with or without OE (the figure below). Therefore, these results indicate that, at least in the GC setting, CLDN18-ARHGAP26 fusion-induced FFA production may trigger immunosuppression through Treg modulation in a ferroptosis-independent manner. Anyway, we greatly appreciate your constructive comments and hope our extra results can answer your concerns.

Referee #2 (Comments on Novelty/Model System for Author):

It is reasonable to detect anti-tumor capacity both in in vitro co-culture models and in vivo xenograft gastric cancer models.

Response: We deeply appreciate your comments regarding the novelty and model system of our study. We have made a point-to-point response to your comments and we hope our revisions can address your concerns.

Referee #2 (Remarks for Author):

This manuscript focuses on gene fusion neoantigen as an effective cancer target, the authors first identified a recurrent fusion gene CLDN18-ARHGAP in Chinese gastric cancer cohorts, then they examined the immunogenic potential through in vitro co-culture models and in vivo xenograft gastric cancer models. Finally, the key mechanism how CLDN18-ARHGAP regulated tumor pathways and tumor immune microenvironment were explored. The design of the study is reasonable and the manuscript is written well, I only have one important concern about how the authors select CLDN18-ARHGAP as their investigation target. The mutation landscape of gastric cancer cohort only include CLDN18-ARHGAP related fusion gene, how about other fusion gene? The authors should clarify it.

Response: Thanks a lot for your extraordinarily comprehensive and detailed comments on our study. Gene fusions represent an important class of somatic alterations in cancer and play a significant role in the initial steps of tumorigenesis [1]. In 2014, The Cancer Genome Atlas (TCGA) first found the occurrence of CLDN18-ARHGAP fusion gene in 4% of GC [2]. And in 2018, Yang et al. identified a frequent proportion of CLDN18-ARHGAP fusion gene up to 25% in gastric signet ring cell carcinoma (SRCC) [3]. Also, CLDN18-ARHGAP gene fusions were detected in 9% (8/87) of GC patients and 19% (5/26) of poorly cohesive GC patients in our patient cohort. Moreover, it is verified that the CLDN18-ARHGAP fusion gene is an oncogenic driver in GC, and patients with CLDN18-ARHGAP fusion were associated with worse survival outcomes and chemotherapy resistance [3]. These findings suggest that CLDN18-ARHGAP gene fusion is a potent target for gastric cancer with high occurrence. However, it remains unknown how it could be targeted. As a result, we chose CLDN18-ARHGAP gene fusion as our investigation target. In our study, we identified the CLDN18-ARHGAP fusion gene as a critical source of immunogenic neoepitopes, a key regulator of the

tumor immune microenvironment, and immunotherapeutic applications specific to this oncogenic fusion. The mutation landscape of gastric cancer cohort in our centre also include SMIM19-KAT6A fusion gene (shown in the source data of Fig. 1A). However, it is single occurrence without significant clinic value. Actually, we also found FGFR2 gene fusion in another GC patient cohort before (the next generation sequencing panel didn't include CLDN18 probe). And, we verified that GC patients with FGFR2 fusion may benefit from FGFR2 inhibitor [4]. We hope our response can answer your concerns

Reference:

- [1] F. Mitelman, B. Johansson, F. Mertens, The impact of translocations and gene fusions on cancer causation, *Nat. Rev. Canc.* 7 (2007) 233–245.
- [2] Comprehensive molecular characterization of gastric adenocarcinoma. *Nature*, 2014. 513(7517): p. 202-9.
- [3] Shu Y, Zhang W, Hou Q, et al., Prognostic significance of frequent CLDN18-ARHGAP26/6 fusion in gastric signet-ring cell cancer. *Nat Commun*, 2018. 9(1): p. 2447.
- [4] Wang Y, Shi T, Wang X, et al., FGFR2 alteration as a potential therapeutic target in poorly cohesive gastric carcinoma. *J Transl Med.* 2021 Sep 22;19(1):401.

Referee #3 (Comments on Novelty/Model System for Author):

This is an interesting study, which is focused on a potential important target of gastric cancers. Although the CLDN18-ARHGAP fusion has been identified and investigate for some years. There is no substantial progress in the treatment of this fusion gene. This study given us a new sight for the treatment of GC patients with CLDN18-ARHGAP fusion.

Response: We sincerely thank you for highly evaluating our study and greatly appreciate the valuable and important concerns you raised for us. We have addressed each concern separately and replied as follows.

Referee #3 (Remarks for Author):

Comment 1. The landscape of mutations should be compared with other studies. Notably, the heatmap does not illustrate any copy number deletion, and the reasons for selecting sCNA genes should be addressed.

Response: We are very grateful for your kind suggestions. TCGA describe a comprehensive molecular evaluation of 295 primary gastric adenocarcinomas. The top mutated genes are TP53 (50%), ARID1A (14%), PIK3CA (12%), CDH1 (11%) and

SMAD4 (8%) [1]. Also, the gastric cancer molecular subtype of the Asian Cancer Research Group (ACRG) was analyzed based on NGS of 300 primary GC tumor specimens. TP53 (33%), ATN1 (25%), MUC6 (22%), ARID1A (18%) and PIK3CA (14%) were the most significantly mutated genes [2]. In our cohort, TP53 (63%), ARID1A (23%), CDH1 (16%), APC (15%), LRP1B (13%), TGFBR2 (13%), FAT3 (11%), PIK3CA (9%), SMARCA4 (9%) and ACVR2A (8%) were the top 10 most frequently mutated SNV&INDEL genes, while CCNE1 (9%), FSR2 (8%) and MDM2 (8%) were the most frequent genes with CNVs in PC GC patients. We can see that TP53, ARID1A, and PIK3CA were the most commonly mutated genes in different populations. We also added the discussion in our manuscript and the corresponding discussion have been revised on page 16 highlighted in yellow.

There are copy number deletions in our cohort, including MLH1, FBXW7, SMAD4, ARHGAP6, ARID2, CDKN2B, FANCC, JAK1, JAZF1 and PRKN (shown in the source data of Fig. 1A). However, all copy number deletions are not recurrent. As a result, we didn't show it in the heatmap. We chose the top5 sCNA genes to display in the heatmap and the new heatmap was replaced in the fig. 1A.

Reference:

[1] Comprehensive molecular characterization of gastric adenocarcinoma. *Nature*, 2014. 513(7517): p. 202-9.

[2] Cristescu, R., et al., Molecular analysis of gastric cancer identifies subtypes associated with distinct clinical outcomes. *Nat Med*. 2015 May;21(5):449-56.

Comment 2. In some experiment designs, three mice per group are not sufficient to fully evaluate the results. At least five mice in each group should be used.

Response: Thanks for pointing out this for us. We have added mice to at least five to fully evaluate the results. The corresponding results have been revised on Fig. 5B. In addition, we also added other in vitro experiments to 5 repeated samples as much as we can (Fig. 5A). We hope our revisions can address your concern.

Comment 3. Although the regulatory phenomenon of CLDN18-ARHGAP on PI3K/Akt signaling has been validated, the underlying regulatory mechanism has not been discussed or investigated.

Response: Thanks a lot for raising this concern for us. ARHGAP family is regulatory

factors of the RHOA pathway. They could stimulate, via their GAP domain, the GTPase activities of RHOA, resulting in their inactivation. Given that the CLDN18-ARHGAP fusion protein retains the GAP domain of ARHGAP, we hypothesized that the fusion may regulate the RHOA pathway [1]. Furthermore, in 2020, Kumagai demonstrated that regulating RHOA could influence the downstream PI3K/AKT signalling pathway [2]. To sum up, we speculated that CLDN18-ARHGAP fusion gene could regulate PI3K/AKT signalling pathway by stimulating RHOA. The corresponding discussion have been revised on page 19 highlighted in yellow. We hope our revisions can address your concern.

References:

- [1] F. Yao, J.P. Kausalya, Y.Y. Sia, et al., Recurrent fusion genes in gastric cancer:CLDN18-ARHGAP26 induces loss of epithelial integrity, *Cell Rep.*, 12 (2015) 272-285.
- [2] S. Kumagai, Y. Togashi, C. Sakai, et al., An oncogenic alteration creates a microenvironment that promotes tumor progression by conferring a metabolic advantage to regulatory T cells, *Immunity*, 53 (2020) 187-203.

Comment 4. Abbreviations should be defined at their first mention in the manuscript.

Response: Thanks for pointing out this for us. We have defined abbreviations at their first mention in the manuscript. The corresponding definitions have been highlighted yellow.

Comment 5. Fig1: log (TMB+1) why plus 1, is there sample carried no mutation? What's the base of log? No rearrangement was observed. What's the definition of short indel and long indel?

Response: Thanks a lot for your in-depth comments. Logarithmic transformation of the TMB can make the data more symmetrical and conform to the assumption of normal distribution for statistical analysis and visualization. However, the results will be infinitesimal when TMB is 0. To ensure no invalid results occur during the calculation, we customized to use log (TMB+1) for analysis. Actually, in this cohort, all patient carried mutation. And, the base of log is e.

We didn't show rearrangements in Fig. 1A as all rearrangements are not recurrent and we have deleted the symbol of rearrangement in Fig. 1A.

Short indel usually refers to small insertion-deletion variants, involving several bases less than 50bp. And, long indel usually refers to large insertion-deletion variants that involve 50 or more bp.

We hope these explanations can answer your questions.

Comment 6. Fig3b: no error bar in the line chart.

Response: We appreciate it a lot for your professional and detailed comments. We totally agree with you that error bars should be added in the line chart. As the amount of PBMCs is limited, we were incapable of doing three repeated samples for each effect/target ratio in previous time. Here, we repeated the experiments and did four repeated samples for effect/target ratio of 10:1. The line chart had been replaced and error bar had been added (Fig. 3B).

Comment 7. There are several typos present in the manuscript that should be corrected. eg: In "loss or amplification of driver genes like PIK3CA, PETN, or AKT": PETN should be PTEN; In "in 2018, Yang et al. identified a frequent proportion of CLDN18-ARHGAP fusion gene up to 25% in gastric signet ring cell carcinoma (SRCC)": Yang et al. should be Shu et al.

Response: Thanks so much for pointing out this for us. We have read the manuscript carefully and corrected the typos. The corresponding corrections have been revised and highlighted in yellow.

Thanks again for your consideration of our revised manuscript.

Sincerely yours,

Jia Wei, M.D., Ph.D.

Department of Oncology, Nanjing Drum Tower Hospital, Affiliated Hospital of Medical School, Nanjing University, Nanjing, China.

Tel: +86-13951785234

Fax: +86-25-83317016

Email: jiawei99@nju.edu.cn

13th Mar 2024

Dear Dr. Wei,

Thank you for the submission of your revised manuscript to EMBO Molecular Medicine. Your manuscript has now been re-reviewed by one of the original reviewers. Based on their advice (included below), I am pleased to inform you that we will be able to accept your manuscript pending the following final amendments:

- 1) Please check the "Author Checklist" carefully and complete all relevant questions. Please add in the section on Human research participants that information is also available in the Materials and Methods section.
- 2) In the main manuscript file, please do the following:
 - Please include up to 5 keywords.
 - Data availability: Please update the NGS dataset to be publicly available, as all deposited datasets should be freely accessible upon publication. Also, please use the following format to report the accession number of your data:
The datasets produced in this study are available in the following databases:
- [data type]: [full name of the resource] [accession number/identifier] ([doi or URL or identifiers.org/DATABASE:ACCESSION])
For example:
The datasets produced in this study are available in the following databases:
- NGS-seq data: Gene Expression Omnibus GSE46748 (<https://www.ncbi.nlm.nih.gov/geo/query/acc.cgi?acc=GSE46748>)
Please check "Author Guidelines" for more information.
<https://www.embopress.org/page/journal/17574684/authorguide#availabilityofpublishedmaterial>
 - Please note that funding information should be removed from the title page and given in the "Acknowledgements" section.
 - Please rename "Conflict of Interest" to "Disclosure and competing interests statement". We updated our journal's competing interests policy in January 2022 and request authors to consider both actual and perceived competing interests. Please review the policy <https://www.embopress.org/competing-interests> and update your competing interests if necessary.
 - Author contributions: Please remove it from the manuscript and only specify author contributions in our submission system. CRediT has replaced the traditional author contributions section because it offers a systematic machine-readable author contributions format that allows for more effective research assessment. You are encouraged to use the free text boxes beneath each contributing author's name to add specific details on the author's contribution. More information is available in our guide to authors:
<https://www.embopress.org/page/journal/17574684/authorguide#authorshippinguidelines>
 - Please correct the reference citation in the reference list - these should be alphabetical, not numerical. Where there are more than 10 authors on a paper, note that only 10 should be listed, followed by "et al.". Please check "Author Guidelines" for more information.
<https://www.embopress.org/page/journal/17574684/authorguide#referencesformat>
 - Data not shown: We do not allow statements/conclusions with "data not shown". As per our guidelines, on "Unpublished Data" the journal does not permit citation of "Data not shown". All data referred to in the paper should be displayed in the main or Expanded View figures. Please remove from page 12.
- 3) In the Materials and Methods, please take care of the following:
 - Cell lines: Please include all information requested in the author checklist for cell lines used in the manuscript - currently clone numbers appear to be missing. Please also be sure to include a sentence in the Materials and Methods as to whether or not the cell lines were recently authenticated and tested for mycoplasma contamination.
 - Please ensure that a statement on whether or not blinding was done is included in the Materials and Methods even if no blinding was done.
 - Antibodies: please ensure that company name, catalog number, and dilutions/amounts of each antibody are reported. Currently some of the information is missing in each of the sections on IFN- γ ELISPOT assay, Flow cytometry analysis, Western blot analysis (please indicate specifically for each antibody whether 1:1000 or 1:2000 dilution was used)
- 4) Please place individual sections of the manuscript in the following order: Title page - Abstract & Keywords - Introduction - Results - Discussion - Materials & Methods - Data Availability - Acknowledgements - Disclosure and Competing Interests Statement - The Paper Explained - For More Information - References - Figure Legends - Tables with legends - Expanded View Figure Legends.
- 5) For the figures and figure legends, please take care of the following:
 - Please remove all figures from main manuscript file and leave only main figure legends placed after the references.
 - Please make sure to update the callouts of all figures in the main manuscript text (currently figure callouts are missing for Figure 7).
 - Please note that a separate 'Data Information' section is required in the legends of figures 3b-c, f-g; 4a-c; 5a-e, g-i; 6b, d-f, h-j.
 - Please note that the legends for figures 6i-j are incorrectly labelled as 6e-f. This needs to be rectified.
 - Please indicate the statistical test used for data analysis in the legends of figures 3b-c, f-g; 4a-c; 5a-e, g-i; 6b, d-f, h-j.
 - Please note that in figures 3b-c, f-g; 4a-c; 6b, d-f, h-j; there is a mismatch between the annotated p values in the figure legend and the annotated p values in the figure file that should be corrected.
 - Please note that information related to n is missing in the legends of figures 2a-b, d-e, g; 3b-c, g; 5a, c-e, g-i; 6e-f, i-j.

- Please note that the error bars are not defined in the legends of figures 2b, d-e, g.

6) Tables: Please change the color scheme from red and green to a different combination of colors (in order to avoid difficulty for people with red-green color blindness) in Table 1.

7) Synopsis:

- Synopsis image: Please provide a synopsis image that summarises the main findings of the manuscript on a glance. Please upload it as a high-resolution jpeg file 550 pixels wide x (250-400) pixels high.

- Synopsis text: Please provide a short standfirst (maximum of 300 characters, including space), limit the bullet points to max. 5 and upload it as a separate .doc file. Please write the bullet points to summarise the key NEW findings. They should be designed to be complementary to the abstract - i.e. not repeat the same text. We encourage inclusion of key acronyms and quantitative information (maximum of 30 words / bullet point). Please use the passive voice.

8) Appendix file: Please upload the Appendix as a single PDF (no separate image files are needed).

9) Source Data: Please zip together all source data files for each figure in a single folder, with the panels clearly visible in the folder structure. Currently there are multiple files for Figure 4.

10) The Paper Explained: Please provide "The Paper Explained" and add it to the main manuscript text. Please check "Author Guidelines" for more information. <https://www.embopress.org/page/journal/17574684/authorguide#researcharticleguide>

11) For more information: This space should be used to list relevant web links for further consultation by our readers. Could you identify some relevant ones and provide such information as well? Some examples are patient associations, relevant databases, OMIM/proteins/genes links, author's websites, etc...

12) As part of the EMBO Publications transparent editorial process initiative (see our policy here:

https://www.embopress.org/transparent-process#Review_Process), EMBO Molecular Medicine will publish online a Peer Review File (PRF) to accompany accepted manuscripts. This file will be published in conjunction with your paper and will include the anonymous referee reports, your point-by-point response and all pertinent correspondence relating to the manuscript. Let us know whether you agree with the publication of the PRF and as here, if you want to remove or not any figures from it prior to publication. Please note that the Authors checklist will be published at the end of the PRF.

13) Please provide a point-by-point letter INCLUDING my comments as well as the reviewer's reports and your detailed responses (as Word file).

I look forward to reading a new revised version of your manuscript as soon as possible.

Yours sincerely,

Poonam Bheda

Poonam Bheda, PhD
Scientific Editor
EMBO Molecular Medicine

*** Instructions to submit your revised manuscript ***

***** Reviewer's comments *****

Referee #1 (Comments on Novelty/Model System for Author):

the authors have fully addressed my concerned re model systems.

Referee #1 (Remarks for Author):

The authors have fully addressed my concerned re model systems.

The authors addressed the editorial issues.

1st Jul 2024

Dear Dr. Wei,

Thank you for the submission of your revised manuscript to EMBO Molecular Medicine. We have a few final formatting requests prior to accepting your manuscript:

It is great news that the NGS data have been publicly released and are available on the GSA website. Please update the Data Availability statement with the link and ensure that the formatting follows this example:

- [data type]: [full name of the resource] [accession number/identifier] ([doi or URL or identifiers.org/DATABASE:ACCESSION])

For example:

- NGS-seq data: Gene Expression Omnibus GSE46748 (<https://www.ncbi.nlm.nih.gov/geo/query/acc.cgi?acc=GSE46748>)

You may also include the references to the GSA as you requested in your email.

As previously requested, please include a sentence in the Materials and Methods as to whether or not the cell lines were recently authenticated.

We apologize that the following was not properly requested from you previously, however, we require exact p-values to be reported either in the figure or the figure legend. Please rectify this for all relevant figure panels.

It is fine that Figure 7 is the same as the synopsis figure. However, for the synopsis figure, we require that the figure fits our specified dimensions of 550 pixels wide and 250-400 pixels high. Currently when we resize the figure to 550 pixels wide, it is too tall. Please modify the synopsis figure to fit our required dimensions.

We recently started to require all research articles to format the Materials and Methods using our 'Structured Methods' format, which is required for all research articles. According to this format, the Methods section includes a Reagents and Tools Table (listing key reagents, experimental models, software and relevant equipment and including their sources and relevant identifiers) followed by a Methods and Protocols section describing the methods using a step-by-step protocol format. The aim is to facilitate adoption of the methodologies across labs. More information on how to adhere to this format as well as a downloadable template (.docx) for the Reagents and Tools Table can be found in our author guidelines: <https://www.embopress.org/page/journal/17574684/authorguide#structuredmethods>

We would request that you please update your manuscript to include these details.

Thank you for submitting a new revised version of your manuscript as soon as possible.

Yours sincerely,

Poonam Bheda

Poonam Bheda, PhD
Scientific Editor
EMBO Molecular Medicine

The authors addressed the remaining editorial issues.

2nd Aug 2024

Dear Dr. Wei,

We are pleased to inform you that your manuscript is accepted for publication and is now being sent to our publisher to be included in the next available issue of EMBO Molecular Medicine.

Yours sincerely,

Poonam Bheda, PhD
Scientific Editor
EMBO Molecular Medicine
